# DKM: Differentiable $k$-Means Clustering Layer for Neural Network Compression

**Minsik Cho**[*]     **Keivan Alizadeh-Vahid**[*]     **Saurabh Adya**     **Mohammad Rastegari**

**Apple**

{minsik, kalizadehvahid, sadya, mrastegari}@apple.com

## Abstract

Deep neural network (DNN) model compression for efficient on-device inference becomes increasingly important to reduce memory requirements and keep user data on-device. To this end, we propose a novel differentiable $k$-means clustering layer (DKM) and its application to train-time weight-clustering for DNN model compression. DKM casts $k$-means clustering as an attention problem and enables joint optimization of the DNN parameters and clustering centroids. Unlike prior works that rely on additional parameters and regularizers, DKM-based compression keeps the original loss function and model architecture fixed. We evaluated DKM-based compression on various DNN models for computer vision and natural language processing (NLP) tasks. Our results demonstrate that DKM delivers superior compression and accuracy trade-off on ImageNet1k and GLUE benchmarks. For example, DKM-based compression can offer 74.5% top-1 ImageNet1k accuracy on ResNet50 with 3.3MB model size (**29.4x** model compression factor). For MobileNet-v1, which is a challenging DNN to compress, DKM delivers 63.9% top-1 ImageNet1k accuracy with 0.72 MB model size (**22.4x** model compression factor). This result is 6.8% higher top-1 accuracy and 33% relatively smaller model size than the current state-of-the-art DNN compression algorithms. DKM also compressed a DistilBERT model by **11.8x** with minimal (1.1%) accuracy loss on GLUE NLP benchmarks.

## 1 Introduction

Deep neural networks (DNN) have demonstrated super-human performance on many cognitive tasks (Silver et al., 2018). While a fully-trained uncompressed DNN is commonly used for server-side inference, on-device inference is preferred to enhance user experience by reducing latency and keeping user data on-device. Many such on-device platforms are battery-powered and resource-constrained, demanding a DNN to meet the stringent resource requirements such as power-consumption, compute budget and storage-overhead (Wang et al., 2019b; Wu et al., 2018).

One solution is to design a more efficient and compact DNN such as MobileNet (Howard et al., 2017) by innovating the network architecture or by leveraging Neural Architecture Search (NAS) methods (Liu et al., 2019; Tan et al., 2019). Another solution is to compress a model with small accuracy degradation so that it takes less storage and reduces System-on-Chip (SoC) memory bandwidth utilization, which can minimize power-consumption and latency. To this end, various DNN compression techniques have been proposed (Wang et al., 2019b; Dong et al., 2020; Park et al., 2018; Rastegari et al., 2016; Fan et al., 2021; Stock et al., 2020; Zhou et al., 2019; Park et al., 2019; Yu et al., 2018; Polino et al., 2018). Among them, weight-clustering/sharing (Han et al., 2016; Wu et al., 2018; Ullrich et al., 2017; Stock et al., 2020) has shown a high DNN compression ratio where weights are clustered into a few shareable weight values (or centroids) based on $k$-means clustering. Once weights are clustered, to shrink the model size, one can store indices (2bits, 4bits, etc. depending on the number of clusters) with a lookup table rather than actual floating-point values.

---

[*]equal contribution

Designing a compact DNN architecture and enabling weight-clustering together could provide the best solution in terms of efficient on-device inference. However, the existing model compression approaches do not usefully compress an already-compact DNN like MobileNet, presumably because the model itself does not have significant redundancy. We conjecture that such limitation comes from the fact that weight-clustering through $k$-means algorithm (both weight-cluster assignment and weight update) has not been fully optimized with the target task. The fundamental complexity in applying $k$-means clustering for weight-sharing comes from the following: **a)** both weights and corresponding k-means centroids are free to move (a general $k$-means clustering with fixed observations is already NP-Hard), **b)** the weight-to-cluster assignment is a discrete process which makes $k$-means clustering non-differentiable, preventing effective optimization.

In this work, we propose a new layer without learnable parameters for differentiable $k$-means clustering, DKM, based on an attention mechanism (Bahdana et al., 2015) to capture the weight and cluster interactions seamlessly, and further apply it to enable train-time weight-clustering for model compression. Our major contributions include the following:

- We propose a novel differentiable $k$-means clustering layer (DKM) for deep learning, which serves as a generic neural layer to develop clustering behavior on input and output.

- We demonstrate that DKM can perform multi-dimensional $k$-means clustering efficiently and can offer a high-quality model for a given compression ratio target.

- We apply DKM to compress a DNN model and demonstrate the state-of-the-art results on both computer vision and natural language models and tasks.

## 2 RELATED WORKS

**Model compression using clustering**: DeepCompression (Han et al., 2016) proposed to apply $k$-means clustering for model compression. DeepCompression initially clusters the weights using $k$-means algorithm. All the weights that belong to the same cluster share the same weight value which is initially the cluster centroid. In the forward-pass, the shared weight is used for each weight. In the backward-pass, the gradient for each shared weight is calculated and used to update the shared value. This approach might degrade model quality because it cannot formulate weight-cluster assignment during gradient back propagation (Yin et al., 2019). ESCQ (Choi et al., 2017; 2020) is optimizing the clusters to minimize the change in the loss by considering hessian. Therefore, it is to preserve the current model state, instead of searching for a fundamentally better model state for compression.

HAQ (Wang et al., 2019b) uses reinforcement learning to search for the optimal quantization policy on different tasks. For model compression, HAQ uses $k$-means clustering similar to DeepCompression yet with flexible bit-width on different layers. Our work is orthogonal to this work because the $k$-means clustering can be replaced with our DKM with a similar flexible configuration. "And The Bit Goes Down" (Stock et al., 2020) algorithm is based on Product Quantization and Knowledge Distillation. It evenly splits the weight vector of $N$ elements into $N/d$ contiguous $d$ dimensional sub-vectors, and clusters the sub-vectors using weighted $k$-means clustering to minimize activation change from that of a teacher network. GOBO (Zadeh et al., 2020) first separates outlier weights far from the average of the weights of each layer and stores them uncompressed while clustering the other weights by an algorithm similar to $k$-means.

**Model compression using regularization**: Directly incorporating $k$-means clustering in the training process is not straightforward (Wu et al., 2018). Hence, (Ullrich et al., 2017) models weight-clustering as Gaussian Mixture Model (GMM) and fits weight distribution into GMM with additional learning parameters using KL divergence (i.e., forcing weight distribution to follow $k$ Gaussian distributions with a slight variance). (Wu et al., 2018) proposed deep $k$-means to enable weight-clustering during re-training. By forcing the weights that have been already clustered to stay around the assigned center, the hard weight-clustering is approximated with additional parameters. Both (Ullrich et al., 2017) and (Wu et al., 2018) leverage regularization to enforce weight-clustering with additional parameters, which will interfere with the original loss target and requires additional updates for the new variables (i.e., singular value decomposition (SVD) in (Wu et al., 2018)). Also, relying on the modified loss cannot capture the dynamic interaction between weight distributions and cluster centroids within a batch, thus requiring an additional training flow for re-training.

**Enhance Model compression using dropout**: Quant-Noise (Fan et al., 2021) is a structured dropout which only quantizes a random subset of weights (using any quantization technique) and thus can improve the predictive power of a compressed model. For example, (Fan et al., 2021) showed good compression vs. accuracy trade-off on ResNet50 for ImageNet1k.

**Model quantization**: Besides clustering and regularization methods, model quantization can also reduce the model size, and training-time quantization techniques have been developed to improve the accuracy of quantized models (Li et al., 2019; Zhao et al., 2019). EWGS (J. Lee, 2021) adjusts gradients by scaling them up or down based on the Hessian approximation for each layer. PROFIT (Park & Yoo, 2020) adopts an iterative process and freezes layers based on the activation instability.

**Efficient networks**: Memory-efficient DNNs include MobileNet (Howard et al., 2017; Sandler et al., 2018), EfficientNet (Tan & Le, 2019; 2021) and ESPNet (Mehta et al., 2019). MobileNet-v1 (Howard et al., 2017) on ImageNet1k dataset has top-1 accuracy of 70.3% with 16.1 MB of memory in comparison to a ResNet18 which has 69.3% accuracy with 44.6 MB of model size. Our method can be applied to these compact networks to reduce their model sizes further.

# 3 ALGORITHM

## 3.1 MOTIVATION

Popular weight-clustering techniques for DNN model compression (J. Lee, 2021; Han et al., 2016; Dong et al., 2020; Stock et al., 2020) are based on $k$-means clustering along with enhancements such as gradient scaling/approximation. Using $k$-means clustering, the weights are clustered and assigned to the nearest centroids which are used for forward/backward-propagation during training as illustrated in Fig. 1 (a). Such conventional methods with clustering have two critical drawbacks:

- The weight-to-cluster assignment in conventional approaches is not optimized through back-propagation of training loss function.
- Gradients for the weights are computed in an ad-hoc fashion: the gradient of a centroid is re-purposed as the gradient of the weights assigned to the centroid.

These limitations are more pronounced for the weights on the boundary such as $i$ and $j$ in Fig. 1 (a). In the conventional approaches, $i$ and $j$ are assigned to the centroids $C_2$ and $C_1$ respectively, simply because of their marginal difference in a distance metric. However, assigning $i$ to $C_0$ and $j$ to $C_2$ could be better for the training loss as their difference in distance is so small (Nagel et al., 2020).

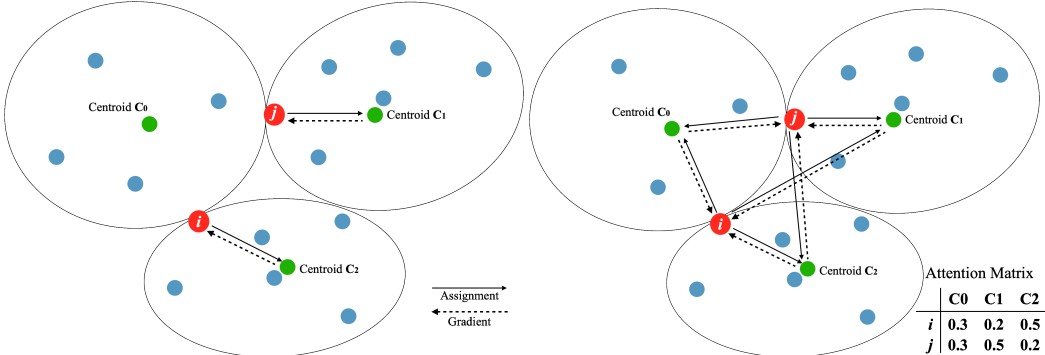

(a) Conventional weight-clustering (Han et al., 2016; Wang et al., 2019b; Stock et al., 2020; J. Lee, 2021)

(b) Attention-based weight-clustering in DKM

**Figure 1:** In conventional weight-clustering algorithms, the boundary weights $i$ and $j$ are assigned to clusters $C_2$ and $C_1$ based on the distance metric respectively, which is neither necessarily suitable for the task nor differentiable against the loss function. DKM instead applies soft assignment using attention mechanism during forward-propagation and enables differentiable backward-propagation, which allows weights to consider other non-nearest clusters (especially helpful for the boundary weights) and shuttle among multiple clusters in order to directly optimize their assignments based on the task loss function.

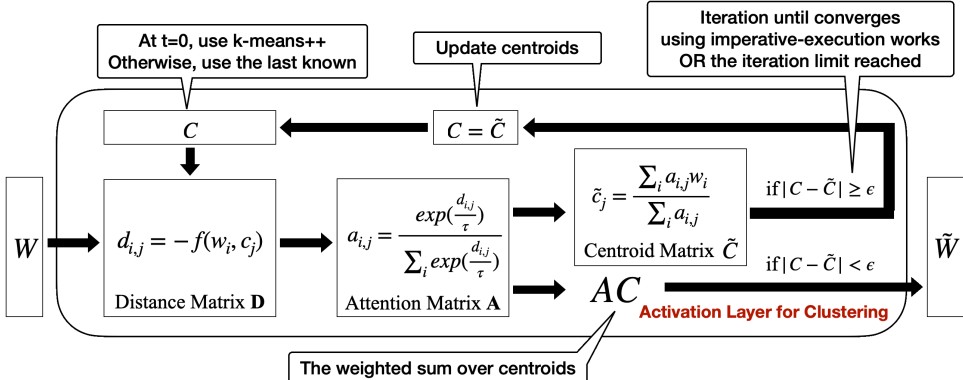

**Figure 2:** Weight-sharing using attention matrix $\mathbf{A}$ is iteratively performed in a DKM layer until the centroids ($C$) converge. Once converged, a compressed weight, $\tilde{W}$ is used for forward-propagation. Since DKM is a differentiable layer, backward-propagation will run through the iterative loop and the gradients for the weights will be computed against the task loss function.

Such lost opportunity cost is especially higher with a smaller number of centroids (or fewer bits for quantization), as each *unfortunate* hard assignment can degrade the training loss significantly.

We overcome such limitations with DKM by interpreting weight-centroid assignment as distance-based attention optimization (Bahdana et al., 2015) as in Fig. 1 (b) and letting each weight interact with all the centroids. Such attention mechanism naturally cast differentiable and iterative $k$-means clustering into a parameter-free layer as in Fig. 2. Therefore, during backward-propagation, attention allows a gradient of a weight to be a product of the attentions and the gradients of centroids, which in turn impact how the clustering and assignment will be done in the next batch. overall our weight assignment will align with the loss function, and can be highly effective for DNN compression.

### 3.2 Differentiable K-means clustering layer for Weight-clustering

DKM can perform a differentiable train-time weight-clustering iteratively for $k$ clusters as shown in Fig. 2 for the DNN model compression purpose. Let $\mathbf{C} \in R^k$ be a vector of cluster centers and $\mathbf{W} \in R^N$ be a vector of the weights, and then DKM performs as follows:

- In the first iteration $\mathbf{C}$ can be initialized either by randomly selected $k$ weights from $\mathbf{W}$ or $k$-means++. For all subsequent iterations, the last known $\mathbf{C}$ from the previous batch is used to accelerate the clustering convergence.

- A distance matrix $\mathbf{D}$ is computed for every pair between a weight $w_i$ and a centroid $c_j$ using a differentiable metric $f$ (i.e., the Euclidean distance) using $d_{ij} = -f(w_i, c_j)$.

- We apply softmax with a temperature $\tau$ on each row of $\mathbf{D}$ to obtain attention matrix $\mathbf{A}$ where $a_{i,j} = \frac{exp(\frac{d_{i,j}}{\tau})}{\sum_k exp(\frac{d_{i,k}}{\tau})}$ represents the attention from $w_i$ and $c_j$.

- Then, we obtain a centroid candidate, $\tilde{\mathbf{C}}$ by gathering all the attentions from $\mathbf{W}$ for each centroid by computing $\tilde{c}_j = \frac{\sum_i a_{i,j} w_i}{\sum_i a_{i,j}}$ and update $\mathbf{C}$ with $\tilde{\mathbf{C}}$ if the iteration continues.

- We repeat this process till $|\mathbf{C} - \tilde{\mathbf{C}}| \leq \epsilon$ at which point $k$-means has converged or the iteration limit reached, and we compute $\mathbf{AC}$ to get $\tilde{\mathbf{W}}$ for forward-propagation (as in Fig. 2).

The iterative process will be dynamically executed imperatively in PyTorch (Paszke et al., 2019) and Tensorflow-Eager (Agrawal et al., 2019) and is differentiable for backward-propagation, as $\tilde{\mathbf{W}}$ is based on the attention between weights and centroids. DKM uses soft weight-cluster assignment which could be hardened in order to impose weight-clustering constraints. The level of hardness can be controlled by the temperature $\tau$ in the softmax operation. During inference we use the last attention matrix (i.e., $\mathbf{A}$ in Fig. 2) from a DKM layer to snap each weight to the closest centroid of the layer and finalize weight-clustering as in prior arts (i.e., no more attention), but such assignment is

expected to be tightly aligned with the loss function, as the weights have been annealed by shuttling among centroids. A theoretical interpretation of DKM is described in Appendix G.

Using DKM for model compression allows a weight to change its cluster assignment during training, but eventually encourages it to settle with the best one w.r.t the task loss. Optimizing both weights and clusters simultaneously and channeling the loss directly to the weight-cluster assignment is by the attention mechanism. Since DKM is without additional learnable parameters and transparent to a model and loss function, we can reuse the existing training flow and hyper-parameters. The key differences between DKM-based compression and the prior works can be summarized as follows:

- Instead of hard weight-cluster assignment and approximated gradient (Han et al., 2016; Wang et al., 2019b; J. Lee, 2021; Stock et al., 2020), DKM uses flexible and differentiable attention-based weight-clustering and computes gradients w.r.t the task loss.

- Instead of modifying the loss function with regularizers to enforce clustering (Ullrich et al., 2017; Wu et al., 2018), DKM can be inserted into forward pass, making the optimization fully aligned with the task objective (i.e., no interference in loss).

- DKM requires no additional learnable parameters (Ullrich et al., 2017; J. Lee, 2021), thus making the training flow simple. For example, DKM-base approach does not need to substitute a convolution layer with a specialized version with additional learning parameters.

- DKM requires no additional computation such as Hessian trace (Dong et al., 2020; Choi et al., 2017) or SVD (J. Lee, 2021; Wu et al., 2018) for gradient approximation, because DKM uses a differentiable process.

- DKM-based compression does not require a complex training techniques such as freezing/progress (Park & Yoo, 2020) or distillation (Stock et al., 2020).

## 3.3 MULTI-DIMENSIONAL DKM

DKM can be naturally extended into multi-dimensional weight-clustering (Stock et al., 2020) due to its simplicity, and is highly effective due to its differentiability. We split $N$ elements of weights into $\frac{N}{d}$ contiguous $d$ dimensional sub-vectors and cluster the sub-vectors ($\mathbf{W} \in R^{\frac{N}{d}*d}$). For example, we simply flatten all the convolutional kernels into a $(\frac{N}{d}, d)$ matrix across both kernel and channel boundaries and apply multi-dimensional DKM to the matrix for clustering in our implementation as in Fig. 3. Accordingly, the cluster centroids will become $d$-dimensional as well ($\mathbf{C} \in R^{k*d}$) and the metric calculation is done in the $d$-dimensional space. With the multi-dimensional scheme, the effective *bit-per-weight* becomes $\frac{b}{d}$ for $b$-bit/$d$-dim clustering. The memory complexity of a DKM layer with $N$ parameters is $O(r\frac{N}{d}2^b)$ where $r$ is the number of iterations per Fig. 2 (i.e., all the intermediate results such as $\mathbf{D}$ and $\mathbf{A}$ at each iteration need to be kept for backward-propagation).

Such multi-dimensional clustering could be ineffective for conventional methods (i.e., DNN training not converging) (Stock et al., 2020; Wang et al., 2019b; J. Lee, 2021), as now a weight might be on the boundary to multiple centroids, and the chance of making wrong decisions grows exponentially with the number of centroids. For example, there are only two centroids for 1bit/1dim clustering, while there are 16 centroids in 4bit/4dim clustering, although both have the same effective *bit-per-weight*. Intuitively, however, DKM can work well with such multi-dimensional configurations as DKM naturally optimizes the assignment w.r.t the task objective and can even recover from a wrong assignment decision over the training-time optimization process.

$\ldots, c_i, c_j, \ldots$

$[w_i, w_j, w_k] \rightarrow$ 0.6, 0.3

$\frac{N}{d}$

$d$

Attention Matrix $\mathbf{A}$

compression over 32bit $\quad cr = \frac{32N}{b\frac{N}{d}} = d\frac{32}{b}$

probability of using $c_i$ $\quad p_i = P[c_i \in W] = \frac{1}{2^b}$

entropy of $\mathbf{W}$ $\quad e_w = \sum_{c_i \in C} -p_i \log_2 p_i$

$\qquad\qquad = 2^b[-\frac{1}{2^b}\log_2\frac{1}{2^b}] = b$

**Figure 3:** Multi-dimensional DKM can increase the weight entropy ($e_w$) with a fixed compression ratio target ($cr$) by increasing a dimension ($d$), which may improve the model accuracy (Park et al., 2017).

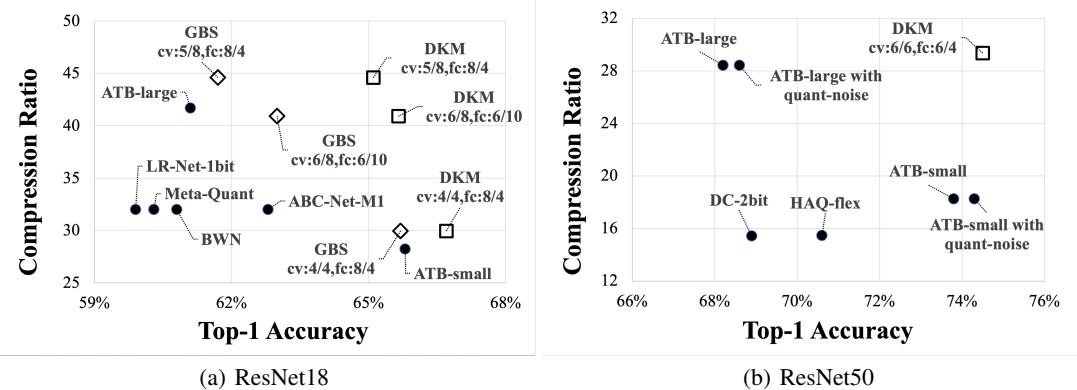

(a) ResNet18                 (b) ResNet50

| Model | Metrics | Base 32bit | DC 2bit | HAQ flex | ATB small | ATB large | DKM configuration | | b/w* |
|---|---|---|---|---|---|---|---|---|---|
| ResNet18 | Top-1 (%) | 69.8 | | | 65.8 | 61.1 | 65.1✓ | cv†:5/8§ | 0.717 |
| | Size (MB) | 44.6 | | | 1.58 | 1.07 | **1.00** | fc‡:8/4 | |
| ResNet50 | Top-1 (%) | 76.1 | 68.9 | 70.6 | 73.8 74.3△ | 68.2 68.8△ | **74.5** | cv:6/6 | 1.077 |
| | Size (MB) | 97.5 | 6.32 | 6.30 | 5.34 | 3.43 | **3.32** | fc:6/4 | |
| MobileNet v1 | Top-1 (%) | 70.9 | 37.6 | 57.1 | nc° | nc | **63.9** | cv:4/4 | 1.427 |
| | Size (MB) | 16.1 | 1.09 | 1.09 | | | **0.72** | fc:4/2 | |
| MobileNet v2 | Top-1 (%) | 71.9 | 58.1 | 66.8 | nc | nc | **68.0** | cv:2/1 | 2.010 |
| | Size (MB) | 13.3 | 0.96 | 0.95 | | | **0.84** | fc:4/4 | |

* effective bit-per-weight (see Section 3.3); ° not converging
† the convolution layers ; ‡ the last fully connected layer;
§ clustering with 6 bits and 8 dimensions; △ ATB with quantization-noise (Fan et al., 2021)
✓ also, 65.7% Top-1 accuracy and 1.09 MB with cv:6/8, fc:6/10
66.7% Top-1 accuracy and 1.49 MB with cv:4/4, fc:8/4

**Figure 4:** Accuracy vs. compression trade-offs: DKM-powered compression (in white box markers) delivers the Pareto superiority to the other schemes (i.e., the top-right corner is the best trade-off) for ResNet18, ResNet50, and MobileNet-v1/v2 on ImageNet1k.

The key benefit of multi-dimensional DKM is captured in Fig. 3. For a given $N \gg d$ in 32 bits, the compression ratio is $d\frac{32}{b}$ (i.e., a function of both $d$ and $b$). Assuming the number of sub-vectors assigned to each centroid is same, the entropy of $\mathbf{W}$ is simply $b$. Since higher entropy in the weight distribution indicates larger learning capacity and better model quality (Park et al., 2017), increasing $b$ and $d$ at the same ratio as much as possible may improve the model quality for a given target compression ratio (see Section 4.1 for results). However, making $b$ and $d$ extremely large to maximize the entropy of $\mathbf{W}$ might be impractical, as the memory overhead grows exponentially with $b$.

## 4 EXPERIMENTAL RESULTS

We compared our DKM-based compression with state-of-the-art quantization or compression schemes on various computer vision and natural language models. To study the trade-off between model compression and accuracy, we disabled activation quantization in every experiment for all approaches, as our main goal is the model compression. All our experiments with DKM were done on two x86 Linux machine with eight NVIDIA V100 GPUs each in a public cloud infrastructure. We used a SGD optimizer with momentum 0.9, and fixed the learning rate at 0.008 (without individual hyper-parameter tuning) for all the experiments for DKM. Each compression scheme starts with publicly available pre-trained models. The $\epsilon$ is set as 1e–4 and the iteration limit is 5.

### 4.1 IMAGENET1K

We compared our DKM-based compression with prior arts: DeepCompression (or **DC**) (Han et al., 2016), **HAQ** (Wang et al., 2019b), and "And The Bit Goes Down" (or **ATB**) (Stock et al., 2020) com-

| | | ResNet18 | ResNet50 | MobileNet-v1 | MobileNet-v2 |
|---|---|---|---|---|---|
| | Base (32 bit) | 69.8 | 76.1 | 70.9 | 71.9 |
| 3 bit | PROFIT | | | 69.6 | 69.6 |
| | EWGS | **70.5** | **76.3** | 64.4 | 64.5 |
| | PROFIT+EWGS | | | 68.6 | 69.5 |
| | DKM | 69.9 | 76.2 | **69.9** | **70.3** |
| 2 bit | PROFIT | | | 63.4 | 61.9 |
| | EWGS | **69.3** | **75.8** | 52.0 | 49.1 |
| | DKM | 68.9 | 75.3 | **66.4** | **66.2** |
| 1 bit | PROFIT | | | nc° | nc |
| | EWGS | 66.6 | 73.8 | 8.5 | 23.0 |
| | DKM 1/1 | 65.0 | 72.1 | 5.9 | 50.8 |
| | DKM 4/4§ | 67.0 | **73.8** | 60.6 | 55.0 |
| | DKM 8/8 | **67.8** | oom□ | **64.3** | **62.4** |
| $\frac{1}{2}$ bit | DKM 4/8 | 62.1 | 70.6 | 46.5 | 34.0 |
| | DKM 8/16 | **65.5** | **72.1** | **59.8** | **58.3** |

° not converging; □ out of memory ; § clustering with 4 bits and 4 dimensions

**Table 1:** When compared with the latest weight quantization algorithms, DKM-based algorithm shows superior Top-1 accuracy when the network is hard to optimize (i.e., MobileNet-v1/v2) or when a low precision is required (1 bit). Further, with multi-dimensional DKM (see Section 3.3), DKM delivers 64.3 % Top-1 accuracy for MobileNet-v1 with the 8/8 configuration which is equivalent to 1 bit-per-weight.

| Metrics | Base (32bit) | RPS | DKM 4/1 | DKM 4/2 | DKM 8/8§ |
|---|---|---|---|---|---|
| Top-1 | 69.8 | 67.9 | **70.9** | 70.3 | 68.5 |
| cr◇ | 1 | 4 | 8 | 16 | 32 |

◇ compression ratio; § clustering with 8 bits and 8 dimensions

**Table 2:** GoogleNet training performance for ImageNet1k: DKM-based compression offered 2x better compression ratio with 3% higher top-1 accuracy than RPS (Wu et al., 2018).

bined with Quantization-noise (Fan et al., 2021), **ABC-Net** (Lin et al., 2017), **BWN** (Rastegari et al., 2016), **LR-Net** (Shayer et al., 2018), and **Meta-Quant** (Chen et al., 2019). We also compared with **GBS** which uses the same flow as DKM except that **Gumbel-softmax** is used to generate stochastic soft-assignment as attention (Jang et al., 2017). In the **GBS** implementation, we iteratively perform drawing to mitigate the large variance problem reported in (Shayer et al., 2018).

We set the mini-batch size 128 per GPU (i.e., global mini-batch size of 2048) and ran for 200 epochs for all DKM cases. Since the public ATB implementation does not include MobileNet-v1/v2 cases (Howard et al., 2017; Sandler et al., 2018), we added the support for these two by following the paper and the existing ResNet18/50 (He et al., 2016) implementations. Instead of using a complex RL technique as in HAQ (Wang et al., 2019b), for DKM experiments, we fixed configurations for all the convolution layers (noted as cv) and the last fully connected layer (noted as fc), except that we applied 8 bit clustering to a layer with fewer than 10,000 parameters.

Using DKM layer, our compression method offers a Pareto superiority to other schemes as visualized in Fig. 4. For ResNet50 and MobileNet-v1/v2, DKM delivered compression configurations that yielded both better accuracy and higher compression ratio than the prior arts. For ResNet18, DKM was able to make a smooth trade-off on accuracy vs. compression, and find Pareto superior configurations to all others: DKM can get 65.1% Top-1 accuracy with 1MB which is superior to ATB-large, 66.8% Top-1 accuracy with 1.49 MB which is superior to ATB-small, and 65.8% Top-1 accuracy with 1.09 MB as a superior balance point. For MobileNet-v1/v2, ATB failed to converge, but DKM outperforms DC and HAQ in terms of both accuracy and size at the same time. For ATB cases, adding quantization noise improves the model accuracy (Fan et al., 2021) only moderately. GBS in fact shows better performance than ATB, but still worse than the proposed method, even after reducing variance through iteration, especially for the high compression configurations. For ResNet18, GBS with the same bit/dimension targets delivered the following Top-1 accuracies ranging from 61.7% to 65.7%. For details, please refer to Appendix F.

We also compared DKM with a well-known regularization-based clustering method on GoolgeNet in Table 2, **RPS** (Wu et al., 2018) which has demonstrated superior performance to another regularization approach (Ullrich et al., 2017). Note that only convolution layers are compressed, following

|  |  | ALBERT | DistilBERT | BERT-tiny | MobileBERT |
|---|---|---|---|---|---|
|  | Base (32 bit) | 90.6 | 88.2 | 78.9 | 89.6 |
| 3 bit | EWGS | 83.3 | 87.6 | 78.3 | 87.8 |
| 3 bit | DKM | **85.1** | **88.2** | **80.0** | **89.0** |
| 2 bit | EWGS | 79.6 | 85.4 | 77.9 | 81.6 |
| 2 bit | DKM | **81.7** | **87.4** | **80.0** | **83.7** |
| 1 bit | EWGS | 62.0 | 60.9 | 74.5 | 60.2 |
| 1 bit | DKM | 79.0 | 82.8 | **77.4** | 69.8 |
| 1 bit | DKM $4/4^\S$ | **80.0** | **84.0** | 77.2 | **78.3** |

§ clustering with 4 bits and 4 dimensions

**Table 3:** Training performance for QNLI: DKM-based scheme outperforms EWGS in compressing various transformed-based architectures. Also, multi-dimensional DKM (see Section 3.3) largely improved the accuracy of MobileBERT with 1 bit-per-weight target using the 4/4 configuration.

|  | Base | GOBO | DKM | DKM |
|---|---|---|---|---|
| Metrics | 32bit | xform$^\dagger$3,emb$^\ddagger$4 | xform4/2$^\S$,emb4 | xform5/2,emb3 |
| Top-1 | 82.4 | **81.3** | **81.3** | **81.3** |
| Size (MB) | 255.4 | 23.9 | 21.8 | **21.5** |

$^\dagger$ the transformer layers ; $^\ddagger$ the embedding layer
$^\S$ clustering with 4 bits and 2 dimensions

**Table 4:** DistillBert training performance for MNLI: DKM-based compression offered 10% smaller model size with the same accuracy target than GOBO.

the setup in RPS (Wu et al., 2018). Table 2 clearly indicates that DKM can allow both much better compression and higher accuracy than RPS even with 1 bit-per-weight.

We also compared our DKM-based algorithm with the latest scalar weight quantization approaches, **PROFIT** (Park & Yoo, 2020) and **EWGS** (J. Lee, 2021) (which have outperformed the prior arts in the low-precision regimes) by running their public codes on our environments with the recommended hyper-parameter sets. Table 1 summarizes our comparison results on ResNet18, ResNet50, and MobileNet-v1/v2 for the ImageNet1k classification task. Following the experimental protocol in (Zhang et al., 2018; J. Lee, 2021; Rastegari et al., 2016), we did not compress the first and last layers for all the experiments in Table 1. It clearly shows that our approach with DKM can provide compression comparable to or better than other approaches, especially for the low-bit/high-compression regimes. We denote clustering with $b$ bits and $d$ dimensions as $b/d$ as it will assign $\frac{b}{d}$ bits in average to each weight, and the number of weight clusters is $2^b$. Especially with multi-dim clustering such as 4/4 or 8/8 bits, our DKM-based compression outperforms other schemes at 1 bit, while PROFIT cannot make training converge for MobileNet-v1/v2. One notable result is 64.3% Top-1 accuracy of MobileNet-v1 with the 8/8 bit configuration (which is 1 bit-equivalent). DKM with 8/16 bits (effectively 0.5 bit per weight) shows degradation from the 8/8 bit configuration, but still retains a good accuracy level. We also tried PROFIT+EWGS as proposed in (J. Lee, 2021), which showed good results on MobileNet-v1/v2 for 3 bits but failed to converge for 2 and 1 bits.

With the overall compression ratio (or bit-per-weight) fixed, our experiments with DKM confirm that a higher $d$ can yield a better quality training result. For the example of MobileNet-v2, DKM 8/16 yielded 24% better top-1 accuracy than DKM 4/8 although both have the same $\frac{1}{2}$ bit-per-weight, and the same trend is observed in other models. However, DKM 8/8 failed to train ResNet50 due to the memory limitation, while DKM 8/16 successfully trained the same model, because the larger dimension (i.e., 8 vs 16) reduces the memory requirement of the attention matrix as discussed in Section 3.3. For additional discussion, please refer to the Appendix A.

## 4.2 GLUE NLP BENCHMARKS

We compared our compression by DKM with **GOBO** (Zadeh et al., 2020) and **EWGS** (J. Lee, 2021) for BERT models on NLP tasks from the GLUE benchmarks (Wang et al., 2019a), QNLI (Question-answering NLI) and MNLI (Multi NLI). We fixed the learning rate as 1e-4 for all the experiments which worked best for EWGS, and all experiments used mini-batch size 64 per GPU (i.e., global mini-batch size of 1024) with the maximum seq-length 128.

| Network | ResNet18 | | | ResNet50 | MobileNet-v1 | MobileNet-v2 |
|---|---|---|---|---|---|---|
| configuration | cv:5/8 fc:8/4 | cv:4/4 fc:8/4 | cv:6/8 fc:6/10 | cv:6/6 fc:6/4 | cv:4/4 fc:4/2 | cv:2/1 fc:4/4 |
| Inference-time | 65.1 | 66.7 | 65.7 | 74.5 | 63.9 | 68.0 |
| Train-time | 66.0 | 67.0 | 66.5 | 74.7 | 65.6 | 68.8 |

**Table 5:** Top-1 Accuracy with Train-time and Inference-time Weights for Fig. 4

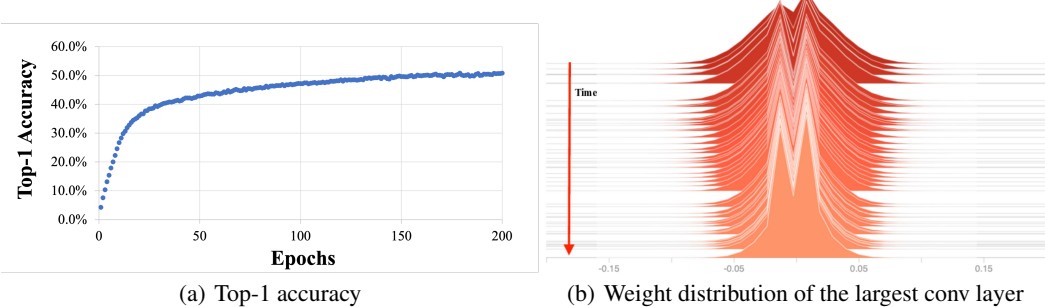

(a) Top-1 accuracy    (b) Weight distribution of the largest conv layer

**Figure 5:** MobileNet-v2 convergence with DKM 1/1: DKM delivers 50.8% top-1 accuracy with 1 bit compression by gradually clustering the weights into two centroids using the task objective only.

We compared our DKM-based compression against EWGS (J. Lee, 2021) on the QNLI dataset, and Table 3 demonstrates that DKM offers better predictability across all the tested models (Lan et al., 2019; Sanh et al., 2019; Turc et al., 2019; Sun et al., 2020) than EWGS. Note that the embedding layers were excluded from compression in QNLI experiments. As in ImageNet1k experiments, the 4/4 bit configuration delivers better qualities than the 1 bit configuration on all four BERT models, and especially performs well for the hard-to-compress MobileBERT. Table 3 also indicates that different transformer architectures will have different levels of accuracy degradation for a given compression target. For the example of 1 bit, MobileBERT degraded most due to many hard-to-compress small layers, yet recovered back to a good accuracy with DKM 4/4.

When DKM compared against GOBO (Zadeh et al., 2020) (which has outperformed the prior arts on BERT compression) on DistilBERT with the MNLI dataset, our results in Table 4 clearly show that DKM offers a better accuracy-compression trade-off than GOBO, and also enables fine-grained balance control between an embedding layer and others: using 2.5 bits for Transformer and 3 bits for embedding is better than 2 bits for Transformer and 4 bits for embedding for DistilBERT.

### 4.3  DKM-BASED COMPRESSION ANALYSIS

Since DKM-based compression uses attention-driven clustering as in Fig. 2 during training but snaps the weights to the nearest centroids, there is a gap between train-time and inference-time weights which in turn leads to accuracy difference between train and validation/test accuracies. Therefore, we measured the Top-1 accuracies with both weights for the DKM cases from Fig. 4 as in Table 5. We observed the accuracy drop is about 0.2% - 1.7% and gets larger with hard-to-compress DNNs.

Fig. 5 shows that our compression approach can offer a smooth convergence and gradual weight-clustering based on task loss back-propagated through DKM layers, without any custom training flow and extra regularization. For additional details and results, please refer to Appendix C.

## 5  CONCLUSION

In this work, we proposed a differentiable $k$-means clustering layer, DKM and its application to model compression. DNN compression powered by DKM yields the state-of-the-art compression quality on popular computer vision and natural language models, and especially highlights its strength in low-precision compression and quantization. The differentiable nature of DKM allows natural expansion to multi-dimensional $k$-means clustering, offering more than 22x model size reduction at 63.9% top-1 accuracy for highly challenging MobileNet-v1.

## 6 REPRODUCIBILITY STATEMENT

Our universal setup for experiments is disclosed in the first paragraph of Section 4 and per-dataset-setups are also stated in the first paragraphs of Sections 4.1 and 4.2 along with key hyper-parameters in Appendix.

For Imagenet1k, we used a standard data augmentation techniques: RandomResizedCrop(224), RandomHorizontalFlip, and Normalize(mean=[0.485, 0.456, 0.406], std=[0.229, 0.224, 0.225]) as in other papers (Park & Yoo, 2020; J. Lee, 2021). During evaluation, we used the following augmentations: Resize(256), CentorCrop(224), and Normalize(mean=[0.485, 0.456, 0.406], std=[0.229, 0.224, 0.225]). For GLUE benchmark, we used the default setup from HuggingFace.

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

## A  ABLATION STUDY: DKM SYSTEM PERFORMANCE AND OVERHEAD

Table 6 shows the GPU memory utilization and per-epoch runtime for the DKM cases from Table 1. While DKM layers have negligible impacts on training speed, the GPU memory utilization increases with more bits (i.e., more clusters) and with smaller dimensions, as discussed in Section 3.3: the memory complexity of a DKM layer with $N$ parameters and $d$ dimension is $O(r\frac{N}{d}2^b)$ where $r$ is the number of iterations.

| | ResNet18 | | ResNet50 | |
| --- | --- | --- | --- | --- |
| | GPU memory utilization (%) | Per-epoch runtime (sec) | GPU memory utilization (%) | Per-epoch runtime (sec) |
| 3 bit | 23.8 | 414.1 | 55.3 | 425.0 |
| 2 bit | 21.7 | 401.2 | 49.2 | 429.3 |
| 1 bit | 20.4 | 413.3 | 45.3 | 426.3 |
| $4/4^\S$ bit | 22.3 | 414.4 | 51.8 | 430.4 |
| $4/8$ bit | 20.7 | 423.6 | 46.7 | 410.4 |
| $8/8$ bit | 51.8 | 409.4 | oom | |
| $8/16$ bit | 35.1 | 421.9 | 78.3 | 454.6 |

$\S$ clustering with 4 bits and 4 dimensions

**Table 6:** Memory and Runtime overheads from DKM on ResNet18/50.

Therefore, one straightforward way to reduce GPU usage and avoid the out-of-memory exception is to limit $r$. In order to overcome the out-of-memory error for ResNet50 with DKM 8/8, we applied a constraint $r < 3$ to all the DKM layers, and could reach the top-1 accuracy of 74.0 % with the 98.2% GPU memory utilization. We believe the fundamental solution to this bottleneck is to use a sparse representation for $\mathbf{A}$ by keeping top-$k$ centroids for each weight which is one of our future works.

## B  ABLATION STUDY: HYPER-PARAMETER $\tau$ SEARCH

In the current DKM implementation, we use a global $\tau$ to control the level of softness in the attention matrix. The selection of $\tau$ affects the model predictive power as shown in Fig. 6 where there appears to be an optimal $\tau$ for a given DNN architecture. For examples of ResNet18/80, $\tau = 2e - 5$ is the best value for the 2 bit clustering.

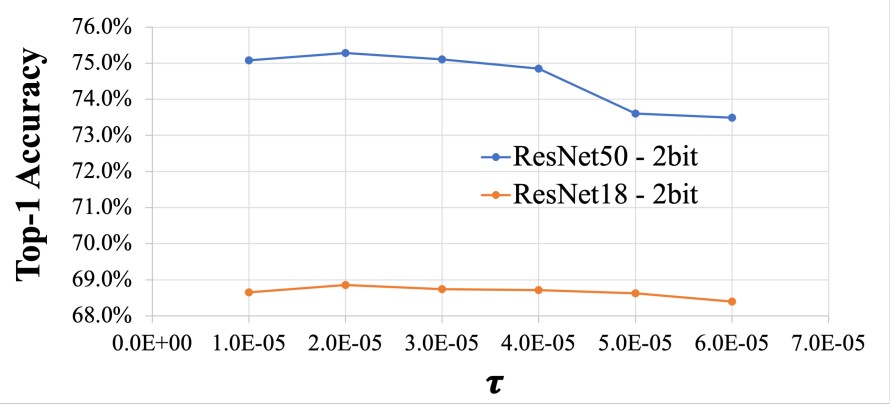

**Figure 6:** ResNet18/50 compression using 2 bits with varying $\tau$ values.

In our experiments, we used a binary search to find out the best $\tau$ values w.r.t. the top-1 accuracy, which are listed in Table 7. In general, one can observe that a complex compression task (i.e., higher

compression targets, more compact networks) tends require a larger $\tau$ to provide enough flexibility or softness.

- For MobileNet-v1/v2, it requires about 10x larger $\tau$ values then for ResNet18/50, because they are based on a more compact architecture and harder to compress.
- When the number of bits decreases, the compression gets harder because there are fewer centroids to utilize, hence requiring a larger $\tau$ value.
- When the centroid dimension increases, the larger $\tau$ value is required, as the compression complexity increases (i.e., need to utilize a longer sequence).

|  | ResNet18 | ResNet50 | MobileNet-v1 | MobileNet-v2 |
|---|---|---|---|---|
| 3 bit | 8.0e-6 | 8.0e-6 | 5.0e-5 | 5.0e-5 |
| 2 bit | 2.0e-5 | 2.0e-5 | 1.0e-4 | 1.0e-4 |
| 1 bit | 5.0e-5 | 5.0e-5 | 3.0e-4 | 1.5e-4 |
| $4/4^\S$ bit | 5.0e-5 | 4.0e-5 | 1.0e-4 | 1.0e-4 |
| 4/8 bit | 5.0e-5 | 5.0e-5 | 1.0e-4 | 1.0e-4 |
| 8/8 bit | 8.0e-5 | oom | 1.0e-4 | 1.0e-4 |
| 8/16 bit | 1.3e-4 | 6.0e-5 | 1.2e-4 | 1.4e-4 |

$\S$ clustering with 4 bits and 4 dimensions

**Table 7:** $\tau$ for the DKM experiments in Table 1 in Section 4

It could be possible to cast $\tau$ as a learnable parameter for each layer or apply some scheduling to improve the model accuracy further (as a future work), but still both approaches need a good initial point which can be found using a binary search technique.

For the BERT experiments with GLUE benchmarks, we used the following $\tau$ regardless of the compression level: 5.0e-5 for ALBERT, 8.0e-5 for DistilBERT, 1.5e-4 for BERT-tiny, and 4.0e-4 for MobileBERT. We found the BERT models are less sensitive to the $\tau$ than ImageNet classifiers.

## C  TRAIN-TIME VS. INFERENCE-TIME WEIGHT DIFFERENCE/ERROR

As discussed in Section 4.3, DKM-based compression requires snap the train-time weights to the nearest centroids for accurate validation or inference, which creates minor accuracy degradation during inference. To understand the behavior better, we measured the Frobenius norm of the weight difference (i.e., $torch.norm(train\_weight - inference\_weight)$) as an error for each layer for every batch from ResNet18 with training with DKM cv:6/8 and fc:6/10 (from Fig. 4) on ImageNet1k. The error changes of the five representative layers for over 120,000 batches or 200 epochs are plotted in Fig. 7 from which we can make the following observations:

- Although every layer starts with a different level of error, the errors get smaller over the training time, and eventually at the scale of 1e–4.
- Aggressive compression makes a layer to begin with a high level of error. For example, the last fc layer starts with an error of 4 (because it targets 0.6 bit per weight).
- the later layers get stabilized better than the earlier layers after enough epochs have been passed.

Our observations are aligned with Fig. 5 (b) in the sense that DKM will encourage weights to be clustered tightly over time, decreasing the difference between train-time and inference-time weights, thus can be very effective in compressing the model and minimizing the model accuracy degradation.

## D  THE NUMBER OF ITERATIONS IN DKM LAYER

In a DKM layer, we iteratively perform $k$-means clustering using the attention mechanism until the clustering process is converged or the maximum iteration number is reached. In our experiments in

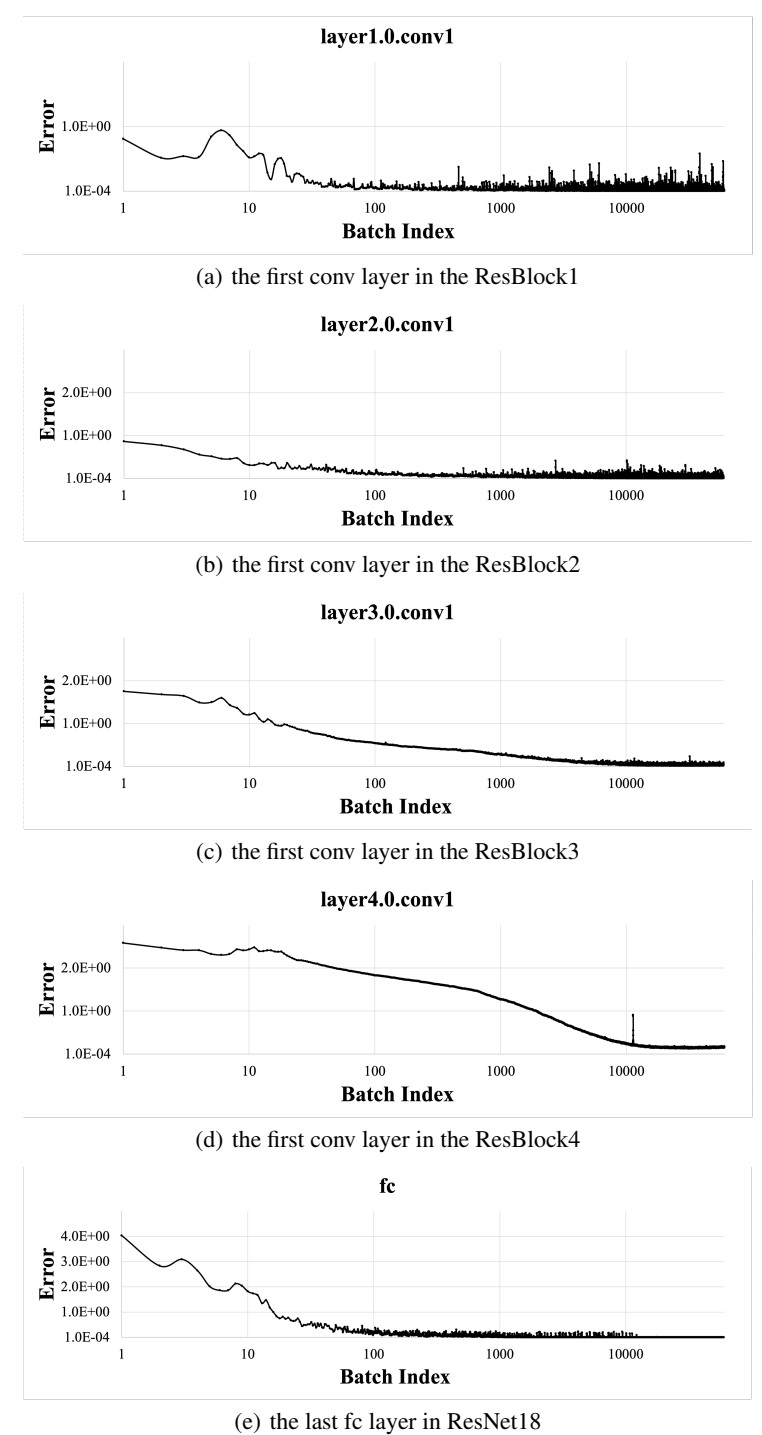

(a) the first conv layer in the ResBlock1

(b) the first conv layer in the ResBlock2

(c) the first conv layer in the ResBlock3

(d) the first conv layer in the ResBlock4

(e) the last fc layer in ResNet18

**Figure 7:** The error between train-time and inference-time weights as a Frobenius norm for every batch for the 200 epochs are plotted, while compressing ResNet18 with DKM cv:6/8 and fc:6/10 on ImageNet1k. After enough number of epochs, such error is reduced to the 1e–4 level.

Section 4, we set the maximum number of iteration as 5 to avoid the out-of-memory error. In order to understand how many iterations are needed before each forward pass, we collected the iteration count per batch from each layer for the first eight epochs (the trend holds true for the remaining epochs) from ResNet18 with training with DKM cv:6/8 and fc:6/10 (from Fig. 4) on ImageNet1k, and plotted the graphs in Fig. 8. Our observations include the following:

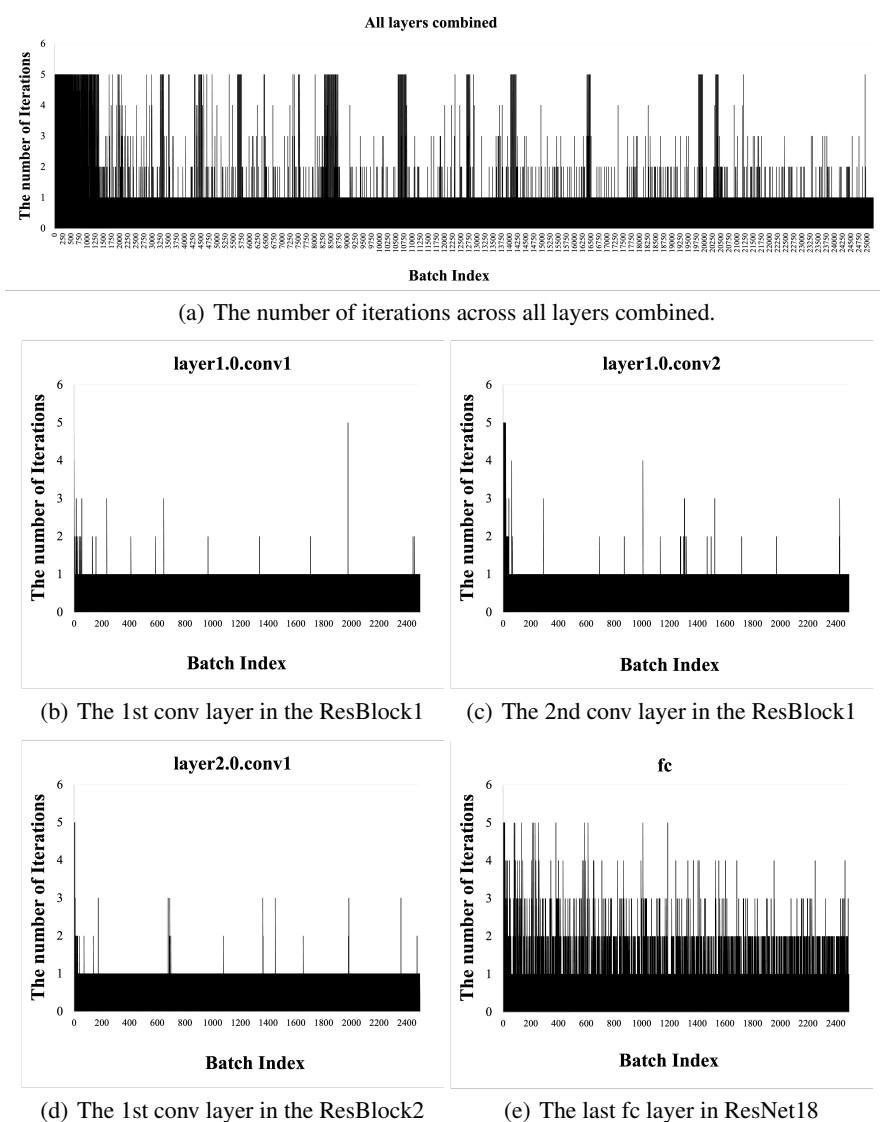

**Figure 8:** The number of iterations in ResNet18 training with DKM cv:6/8 and fc:6/10 on ImageNet1k.

- (a) shows the number of iteration changes over the 8 epochs across all the compressed layers. As one can see, the number of iterations hits the maximum limit initially, implying that the weights are being clustered aggressively.

- As the training continues however, the number of iterations decreases slowly with sporadic spikes, implying DKM-based compression helps learn good weight sharing.

- While the convolutional layers in (b), (c), and (d) get stabilized after only a few dozens of batches, the last fc layer in (e) requires much more iterations throughout the training. We partially believe that this is because we used the DKM 6/10 configuration which is more challenging than the DKM 6/8 for the convolution layer.

## E  $\epsilon$ IN DKM

The $\epsilon$ in Fig. 2 determines when to exit the iterative process. Note that the default value for the experiments in Section 4 is 1e–4 based on from sklearn.cluster.KMeans (https://scikit-learn.org/stable/modules/generated/sklearn.cluster.KMeans.html). Therefore, we performed the sen-

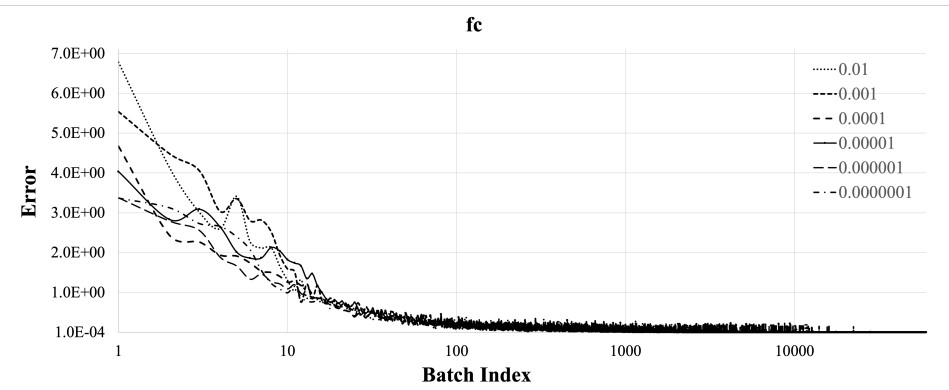

**Figure 9:** We plotted the train vs. inference weight error from the last fc layer of ResNet18 with DKM cv:6/8 and fc:6/10 on ImageNet1k with varying $\epsilon$ from 1e–2 to 1e–7 for 200 epochs. Although the initial error is larger with bigger $\epsilon$, the final errors all converged to a similar level after a sufficient number of epochs.

sitivity analysis by varying the $\epsilon$ from 1e–2 to 1e–7 for ResNet18 with DKM cv:6/8 and fc:6/10 on ImageNet1k and increase the maximum iteration limit to 15 not to make it as a bottleneck. From the results, we found that the final Top-1 accuracies for all the cases were in the range of 65.7 -65.9%. To understand why and develop insights, we plotted the train-time vs. inference-time weight difference of the fc layer in Fig. 9 as similarly with Fig. 7 (d). From the Fig. 9, we could make the following observations:

- With a large $\epsilon$, there is a wider gap between training and inference weights, implying the clustering is not fully optimized. For example, the initial error with $\epsilon = $ 1e–2 is about 2x larger than one with $\epsilon = $ 1e–6.

- However, the gap is closing over time, and after a sufficient number of epochs, the errors are all in the similar range of 1e–4 regardless of the $\epsilon$ value.

- Larger $\epsilon$ values make DKM layers iterate fewer, decreasing the peak memory consumption. For example, the iteration count was 1 throughout training when $\epsilon = $ 1e–2.

Therefore, apparently, as long as the training with DKM layers can run long enough, the selection of $\epsilon$ might not affect the final result. We believe this is because DKM ensures the clustering continuity by resuming from the last known centroid (i.e., from the previous batch). In case that the planned training time is short, a smaller $\epsilon$ value would be preferred but at the cost of larger memory requirement.

## F    GUMBEL-SOFTMAX FOR SOFT ASSIGNMENT

Gumbel-Softmax distribution is a continuous distribution that approximates samples from a categorical distribution and also works with back-propagation. Therefore, it is feasible to use the drawing from Gumbel-softmax as soft assignment to generate attention. Therefore, we experimented Gumbel-Softmax-based DKM (**GBS**) along with the hard assignment scheme (**Hard**) with ResNet18 on ImageNet1k. We ran clustering iteratively for both GBS and Hard, and such iteration helps GBS reduce the variance (Shayer et al., 2018).

We applied the same compression configurations from Fig. 4 and kept all hyper-parameters and training flow intact. The comparison results are in Fig. 10 which shows that Hard is much worse than both GBS and DKM on all the cases with over 10% drop in Top-1 accuracy, proving that soft assignment is a superior way of clustering weights for model compression. Although GBS outperforms Hard, GBS is still worse than DKM on all cases, and the degradation can be as significant as 3.4% drop in Top-1 accuracy for the most aggressive compression target.

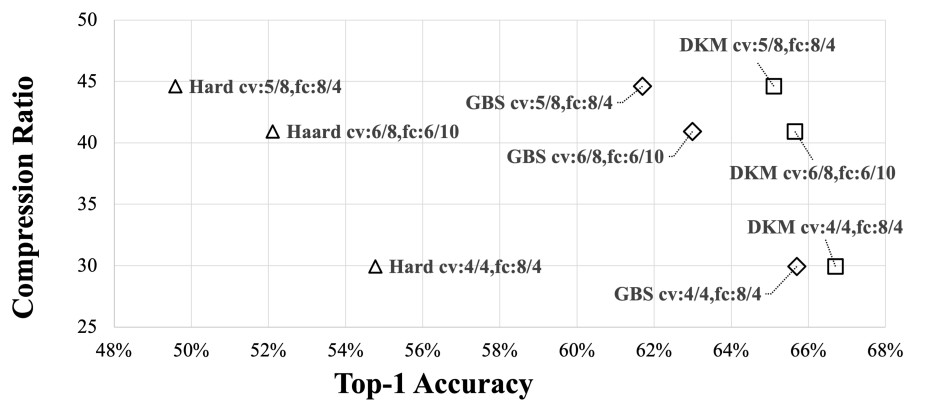

| configuration | cv:4/4,fc:8/4 | cv:6/8,fc:6/10 | cv:5/8,fc:8/4 |
|---|---|---|---|
| DKM | 66.7 | 65.7 | 65.1 |
| GBS | 65.7 | 62.9 | 61.7 |
| Hard | 54.9 | 52.1 | 49.4 |

**Figure 10:** When DKM is compared with another soft assignment method, **GBS** based on Gumbel-softmax and a hard assignment method, **Hard** with ResNet18 on ImageNet1k, DKM outperforms both. Soft assignment techniques outperform hard by a wide margin, and DKM is superior to GBS.

## G RELATION TO EXPECTATION-MAXIMIZATION (EM) AND THEORETICAL INTERPRETATION

Special sub-case of DKM where gradients are not propagated can be related to a standard EM formulation. Here is the formulation-level correspondence: Suppose

$$p(x) = \sum_{i=1}^{K} \frac{1}{K} \mathcal{N}(x|c_i, \sigma^2 = \tau/2)$$

is the Gaussian mixture model over cluster centers. Referring to Fig. 2:

- Maximizing log likelihood of weights

$$\ln P(W|C) = \sum_{i=1}^{N} \ln \Big\{ \sum_{j=1}^{K} \frac{1}{K} \mathcal{N}(w_i|c_j, \sigma^2 = \tau/2) \Big\}$$

  using the EM algorithm is equivalent to DKM for the case of $d_{i,j} = -(w_i - c_j)^2$.

- The attention matrix $\mathbf{A}$ where $a_{i,j} = \frac{exp(\frac{d_{i,j}}{\tau})}{\sum_k exp(\frac{d_{i,k}}{\tau})}$ is equivalent to the responsibilities calculated in the E step: $r_{i,j} = \frac{\frac{1}{K}\mathcal{N}(w_i|c_j, \sigma^2 = \tau/2)}{\sum_{k=1}^{K} \frac{1}{K}\mathcal{N}(w_i|c_k, \sigma^2 = \tau/2)} = a_{i,j}$

- Updating $\mathbf{C}$ using $c_j = \frac{\sum_i a_{i,j} w_i}{\sum_i a_{i,j}}$ is equivalent to M step of EM algorithm. Notice that variance is fixed, therefore M step in EM is only updating cluster centers as DKM does.

However, unlike EM where finding $\mathbf{C}$ every M step is the objective, DKM focuses on generating a representative $\tilde{\mathbf{W}}$ for the train-time compression for DNN.

Even though there is formulation-level similarity between DKM and EM, the way both are optimized is significantly different. While EM iteratively optimizes a specific likelihood function for a set of **fixed** observations, DKM needs to adjust (i.e., optimize) the observations (which are weights) without leading to a trivial solution such as all observations collapsing to a certain point. Hence, DKM can neither assume any statistical distribution nor optimize a specific likelihood function (i.e.,

the observations are dynamically changing). Therefore, DKM uses a simple softmax and rides on the back-propagation to fine-tune the observations w.r.t. the task loss function after unrolling multiple attention updates. When we propagate gradients, then this will turn into a stochastic non-convex joint optimization where we simultaneously optimize observations and centroids for the task loss function, which is shown to offer better accuracy vs. compression trade-offs according to our experiments.

