# OpenReview forum: "DKM: Differentiable k-Means Clustering Layer for Neural Network Compression"
_ICLR.cc/2022/Conference — ICLR 2022 Poster_

### Official Review · Reviewer_5RxN · 2021-10-20

**Correctness:** 2
**Technical Novelty And Significance:** 2
**Empirical Novelty And Significance:** 2
**Recommendation:** 6
**Confidence:** 4

**Main Review:**

This paper proposes a novel differentiable k-means clustering layer (DKM) for deep neural network model compression. The DKM utilizes attention mechanism to align the weight-to-cluster assignment with the training loss function. Overall, the idea is novel but the paper is not prepared enough.
The comments are listed as following:

1.	In Figure 2, there are some confusions in logic. To my understanding, the criterion of convergence should be put after the calculation of Attention Matrix   to decide whether to continue update the Centroid Matrix   or not. However, it was put after updating the Centroid Matrix and weight approximation.
2.	The description of multi-dimensional weight-clustering is not so clear. It is better to help to understand by adding a figure with more details.
3.	In Section 3.2, the procedure of DKM is elaborated. However, theoretical interpretation for each step of DKM is also required.
4.	Some details should be noticed, please check and correct them:
a)	The format of reference should be unified;
b)	Some format errors and spelling mistakes exist in the paper;
c)	The Figure 1 and Figure 2 are not clear, I recommend to use vector graphics;
5.	In Section 4, more related clustering-based quantization methods should be added for further improving the reliability of experiments.


**Summary Of The Paper:**

This paper proposes a novel differentiable k-means clustering layer (DKM) for deep neural network model compression. The DKM utilizes attention mechanism to align the weight-to-cluster assignment with the training loss function. Overall, the idea is novel but the paper is not prepared enough.

**Summary Of The Review:**

This paper proposes a novel differentiable k-means clustering layer (DKM) for deep neural network model compression. The DKM utilizes attention mechanism to align the weight-to-cluster assignment with the training loss function. Overall, the idea is novel but the paper is not prepared enough.
The comments are listed as following:

1.	In Figure 2, there are some confusions in logic. To my understanding, the criterion of convergence should be put after the calculation of Attention Matrix   to decide whether to continue update the Centroid Matrix   or not. However, it was put after updating the Centroid Matrix and weight approximation.
2.	The description of multi-dimensional weight-clustering is not so clear. It is better to help to understand by adding a figure with more details.
3.	In Section 3.2, the procedure of DKM is elaborated. However, theoretical interpretation for each step of DKM is also required.
4.	Some details should be noticed, please check and correct them:
a)	The format of reference should be unified;
b)	Some format errors and spelling mistakes exist in the paper;
c)	The Figure 1 and Figure 2 are not clear, I recommend to use vector graphics;
5.	In Section 4, more related clustering-based quantization methods should be added for further improving the reliability of experiments.

---

> ### Author Response · Authors · 2021-11-17
> **Response to Reviewer3**
>
> We appreciate your valuable reviews. Based on your reviews, we make our theoretical connection to Expectation-Maximization clearer in Section 3 and Appendix G, to provide good insights to readers.
>
> **Q0: In Figure 2, there are some confusions in logic. To my understanding, the criterion of convergence should be put after the calculation of Attention Matrix to decide whether to continue update the Centroid Matrix or not. However, it was put after updating the Centroid Matrix and weight approximation.**
>
> **A0:** Thank you for the comment. The reviewer is right that the criterion of convergence should be put after the calculation of Attention Matrix to decide whether to continue to update the Centroid Matrix or not, and this is exactly what DKM does. In Fig. 2,
> $\hat C$ is temporary and used to decide whether we will update $C$ with $\hat C$: if $|C-\hat C|$ is greater than $\epsilon$, then update does happen. To avoid confusion and make it clear, we notated where the centroid update occurs in Fig. 2.
>
>
> **Q1: The description of multi-dimensional weight-clustering is not so clear. It is better to help to understand by adding a figure with more details.**
>
> **A1:** [cross reference **A3:** for Reviewer0] Thanks for the suggestion. We revised Section 3 and added Fig. 3 to explain the idea better. We hope this would address your concern.
>
> **Q2: In Section 3.2, the procedure of DKM is elaborated. However, theoretical interpretation for each step of DKM is also required.**
>
> **A2:** [cross reference **A0:** for Reviewer2] Thanks for bringing this to our attention. We had described the theoretical interpretation of each step in relation to EM algorithm in Appendix G. We made a reference to the Appendix G in section 3.2. We also added a bulleted list in the Appendix G to track interpretation of each step easier. (Add equations to bullet 2,3).
>
> In short, DKM form itself (without gradients backpropagated) is closely related to a standard EM in terms of formulation. Both have 2 steps in a iterative fashion: **a)** expectation computation vs. attention matrix **b)** maximization step vs. centroid update. Even though DKM shares formulation structural similarity with EM, the way both are optimized is significantly different.
> Instead of optimizing a specific likelihood function for a set of **fixed observations**, we turn this into a stochastic non-convex joint optimization where we simultaneously optimize observations (which is weight in the deep learnig context) and centroids for the task loss function using backward propagation. See Appendix G for in-depth details.
>
>
> **Q3: Some details should be noticed, please check and correct them: a) The format of reference should be unified; b) Some format errors and spelling mistakes exist in the paper; c) The Figure 1 and Figure 2 are not clear, I recommend to use vector graphics;**
>
> **A3:** Thank you for pointing out these. We have brushed up our reference format and gone through the paper to fix cosmetic errors. We also re-gereanted Fig. 1 and 2 to make them more readable.
>
> **Q4: In Section 4, more related clustering-based quantization methods should be added for further improving the reliability of experiments.**
>
> **A4:** [cross reference to **A1:** for Reviewer1] Thank you for the comment. We made more comparisons with the following method to improve the reliability of our experiments.
>
> - Using Gumbel-softmax for soft assignment: We replaced our attention with Gumbel-softmax layer for clustering. See Fig.4 (a) in Section 4 and Fig.10 in Appendix F.
> - Using hard assignment: We used the hard assignment instead of soft assignment, see Fig.10 in Appendix F.
> - Comparison with other methods including LR-Net, BWN, Meta-Quant, ABC-Net, see Fig.4 (a) in Section 4.

---

### Official Review · Reviewer_JTKT · 2021-10-28

**Correctness:** 3
**Technical Novelty And Significance:** 2
**Empirical Novelty And Significance:** 3
**Recommendation:** 6
**Confidence:** 4

**Main Review:**

The paper is mostly well written. The experiments on benchmarks are strong and it is reported that architectures that are already resource-efficient (e.g., MobileNet) can be further compressed using the proposed method.

My main concern is that the paper does not present many insights into the method itself besides good results on benchmark datasets. It is stated that the clustering is adopted to the loss function and not just based on a metric. However, when looking at Figure 2, it appears that the weights are clustered only based on a distance metric. It is stated that it is expected that the weight assignments align well with the loss function. I would require some more intuition, proofs, or experimental evidence to be convinced that this can be expected.

Since the cluster centers are not stored explicitly, they are implicitly stored in the set of weights. Is there some intuition about how the proposed method affects those weights and, therefore, also the implicitly stored cluster centers? Do the weights---before they enter the k-means block---develop some kind of clustering behavior? Or is it just that the weights after the k-means block exhibit clustering.

How confident (i.e., how close to one) are the weight assignments (i.e., attention) to different cluster centers during training? If a weight gets assigned to multiple clusters, I would expect that the weight used for testing might be quite different than the soft weight assigned during training. How does the accuracy change when going from the soft clustered weights (as used during training) to the hard clustered weights (as used during testing).

How important is the initialization of the weights using a pre-trained model? How well does the proposed method work when trained from scratch?

I understand that the proposed method aims to reduce the memory footprint. What about the computational overhead at test time? Does the weight sharing induce some computational overhead at test time? Or can the weight sharing even be exploited to compute predictions faster?

The stopping criterion of k-means appears to be determined by a threshold parameter \epsilon. How critical is this parameter? How often is k-means iterated on average? Is there a discrepancy in the number of iterations of k-means for different layers? Wouldn't it be better to use a fixed number of iterations to ensure a similar behavior throughout the training and to ensure a fixed memory overhead during training?

Minor:
- The word "regress" was used several times which I confused with "regression" (i.e., predicting a real-valued target)
- Why is DKM sometimes referred to as an "activation layer" (e.g., right above Section 3.2). This also confused me since there are weights involved, not activations.

**Summary Of The Paper:**

The paper is concerned with reducing the size of deep neural networks using weight sharing. The paper proposes a new building block that performs a soft k-means algorithm where each weight is assigned a convex combination of the cluster centers. At test-time, the weights are assigned to their closest cluster center such that a real (hard) weight sharing is obtained. The method achieves state-of-the-art accuracy using various architectures on several tasks.

**Summary Of The Review:**

Mostly well-written paper with good experimental results on benchmark data, but hardly any insights into the method itself.

---

> ### Author Response · Authors · 2021-11-17
> **Responses to Reviewer2**
>
> Thank you for your valuable comments. We especially appreciate your feedback on train-time vs test-time weight. It is indeed a very critical point that can strengthen our value proposition, not to mention that it will help readers see through the core mechanism in our work.
>
> **Q0: My main concern is that the paper does not present many insights into the method itself besides good results on benchmark datasets. It is stated that the clustering is adopted to the loss function and not just based on a metric. However, when looking at Figure 2, it appears that the weights are clustered only based on a distance metric. It is stated that it is expected that the weight assignments align well with the loss function. I would require some more intuition, proofs, or experimental evidence to be convinced that this can be expected.**
>
> **A0:** [cross reference **A2:** for Reviewer3] Thanks for the comment. The reviewer is right that weights are clustered by distance in the forward pass. However, when it does weight update after backward pass, the gradient for the weight is (roughly) the product of attention (based on distance) and the **gradient of a centroid w.r.t the task loss**, thus ensuring that the gradient for the weight is also w.r.t. the task loss as well. For example, the gradient $g_i$ for $w_i$ in each iteration would be $g_i = \sum g_c * attention_{i,c} $, where $g_c$ is the gradient of a centroid $c$ as in Fig 1. (b). Therefore, such an update will impact how the clustering and assignment would be done in the next batch, and overall our weight assignment will gradually align with the loss function over many batches. We added intuitive explanations to Section 3.1 for better understanding.
>
> Our method has a theoretical connection to Expectation-Maximization or EM (please refer to Appendix G). While EM finds centroids using fixed observations (or weight), we solve the EM using the back-propagation from the gradient of the loss to optimize the observations (or weight): the conventional EM iteratively finds centroids for given fixed observations. One way to understand the critical difference is following: the observation in EM is not a variable but a static constant, which is no longer valid for deep learning. Thus, in our DKM, we take the weight as optimizable variables, and cast it as an activation layer to get the clustered centroid as output. This is the very first time that an activation layer with clustering behavior is proposed to our best knowledge.
>
> **Q1: Since the cluster centers are not stored explicitly, they are implicitly stored in the set of weights. Is there some intuition about how the proposed method affects those weights and, therefore, also the implicitly stored cluster centers?**
>
> **A1:** Thank you for the opportunity to clarify our method. In fact, we store the last cluster centroids. As in Fig. 2, we use the last known centroids for continuity, as recovering it implicitly can make two successive batches disconnected in terms of the centroid continuity, not to mention it will add computational overhead.
>
>
> **Q2: Do the weights---before they enter the k-means block---develop some kind of clustering behavior? Or is it just that the weights after the k-means block exhibit clustering.**
>
> **A2:** Over time, the weights indeed develop the clustering behavior as shown in Fig 5 (b) where 2 centroids are formed gradually (still moving a bit every iteration) and weights are gathering around either centroid. Since DKM allows forward/backward pass, the gradients for the weights will reflect the clustering effect and let weights adapt to it. **Therefore, both the weights before DKM and after DKM layer will exhibit clustering but the output could have more pronounced clustering, especially in the early stage of training.** In detail, such clustering will get more pronounced as training continues (becoming closer to hard assignment), but will never be perfect hard assignment because DKM uses soft-assignment. For example, the attention of a weight can progress (0.51, 0.49) -> (0.45, 0.55) -> (0.58, 0.32) -> .. -> (0.05, 0.95), yet it never becomes (0, 1). In short, we will get a high-quality *guess* on *right* clustering, but not a clean hard one. This is why we just snap it to the nearest centroid and make it (0,1) for inference/test.

---

> > ### Author Response · Authors · 2021-11-17
> > **Responses to Reviewer2 [continue]**
> >
> > **Q3: How confident (i.e., how close to one) are the weight assignments (i.e., attention) to different cluster centers during training? If a weight gets assigned to multiple clusters, I would expect that the weight used for testing might be quite different than the soft weight assigned during training. How does the accuracy change when going from the soft clustered weights (as used during training) to the hard clustered weights (as used during testing).**
> >
> > **A3:** Thanks for the question. We believe this is related to **Q2**. As pointed out by the reviewer, such snapping creates a gap between train-time and inference-time weights. To understand this better, we measured the difference between the two weights (i.e., torch.norm(train_weight - test_weight) ) layer-by-layer for every batch for ResNet18 and plotted results in Fig. 7 in Appendix C, which clearly shows the error keeps decreasing to a very small enough level over about 120000 batches (or 200 epochs) so that we can rely on our soft assignment method for compression. Table 5 is also added to Section 4 to get the final Top-1 accuracy difference between train-time vs. inference-time weights on Resnet18/50 and Mobilenet v1/v2, which double confirms the accuracy drop is rather limited and the difference at the end is small enough for the proposed compression scheme. We can clearly see that there is degradation, ranging from 0.2% - 1.7%: more degradation for the hard-to-compress networks and for higher compression target. This indicates that our soft assignment causes the gap between train vs. inference weights, yet enables very efficient train-time compression for the SOTA result. For a detailed discussion, please refer to Appendix C.
> >
> > | Network | ResNet18 | ResNet18 | ResNet18 | ResNet50 | MobileNet-v1 | MobileNet-v2 |
> > | --- | --- | --- | --- | -- | --- | -- |
> > | configuration | cv:5/8 | cv:4/4 | cv:6/8 | cv:6/6 | cv:4/4 | cv:2/1 |
> > | | fc:8/4 | fc:8/4 | fc:6/10 | fc:6/4 | fc:4/2 | fc:4/4 |
> > | Inference-time weight | 65.1 | 66.7 | 65.7 | 74.5 | 63.9 | 68.0 |
> > | Train-time weight | 66.0 | 67.0 | 66.5 | 74.7 | 65.6 | 68.8 |
> >
> > **Q4: How important is the initialization of the weights using a pre-trained model? How well does the proposed method work when trained from scratch?**
> >
> > **A4:** Thanks for the question. We have trained ResNet18 with the DKM cv6/8+fc6/10 config from scratch (using Pytorch default initialization), and it yielded only **25.6%**, which implies that the quality of a pre-trained model is critical. We conjecture that the initial clustering is not good enough with the random initialization to converge to a high-accuracy target.
> >
> > **Q5: I understand that the proposed method aims to reduce the memory footprint. What about the computational overhead at test time? Does the weight sharing induce some computational overhead at test time? Or can the weight sharing even be exploited to compute predictions faster?**
> >
> > **A5:** Thank you for the insightful questions. Although DKM layers will be dropped at test time (only centroids and clustering assignments will be kept), there will indeed be computational overhead as the inference now requires lookup (index to weight mapping) to reconstruct the weights. Hence, weight-sharing itself wouldn't make predictions any faster. However, with enough compression, in fact weight-sharing can speed up inference indirectly when moving the weights to a compute-core is relatively more expensive than the compute itself, or when the entire compressed weights can be kept in a fast memory like L1/L2 cache of CPU. Hence, in that sense, it could be possible to target a particular compression ratio to take advantage of the underlying HW.
> >
> > **Q6: The stopping criterion of k-means appears to be determined by a threshold parameter \epsilon. How critical is this parameter?**
> >
> > **A6:** Thank you for the interesting question. We did another experiment with various $\epsilon$ values (from 1e-2 to 1-e7) for Renset18 training.  When we computed the final Top-1 accuracies for all the cases, they all were in the range of 65.7-65.9%. To understand why, we plotted the last FC layer error between train-time vs. inference-time weights (similar to **A3**) in Fig. 9 in Appendix E. With a large $\epsilon$, there is a wider gap between training and inference weights, implying clustering is not fully optimized: the initial error is about 2x larger with$ \epsilon$= 1e−2 than one with $\epsilon$= 1e−6. However, the gap is closing over time, and after many batches, the errors are in a similar range. We believe this is because DKM ensures the clustering continuity by resuming from the last known centroid (i.e., from the previous batch), which allows even training with a larger $\epsilon$ to converge properly after many epochs. Therefore, as long as the training for compression can run long enough, the selection of $\epsilon$ would not affect the final result much. For short training, a smaller $\epsilon$ would be preferred but at a larger memory cost.

---

> > > ### Author Response · Authors · 2021-11-17
> > > **Responses to Reviewer2 [continue again]**
> > >
> > >
> > > **Q7: How often is k-means iterated on average? Is there a discrepancy in the number of iterations of k-means for different layers? Wouldn't it be better to use a fixed number of iterations to ensure a similar behavior throughout the training and to ensure a fixed memory overhead during training?**
> > >
> > > **A7:** [cross reference to **A2:** for Reviewer0] Thank you for the question. First of all, we limit the number of iterations to ensure a fixed memory overhead (i.e., max 5 in our experiments). We further counted the number of iterations for the training of ResNet18 with cv:6/8 fc:6/10 (from Fig. 4), and plotted the results in Fig.8 in Appendix D (for the first 9 epochs only as it is mostly 1 after that). First, it shows that the overall number of iterations decreases in general but at different speeds at different layers. Most notably, while convolutional layers wind down the iterations fast, the last FC layer needs more iterations consistently (although it also needs fewer and fewer iterations as training progresses). We don't have clear reasoning on why, but we hypothesize that it correlates to the magnitude of gradients. Since it needs only a single iteration most of the time, the average iteration number over 200 epochs is 1.0008 exactly for across all the layers.
> > >
> > > **Q8: Minor: The word "regress" was used several times which I confused with "regression" (i.e., predicting a real-valued target)**
> > >
> > > **A8:** Thank you. We replaced 'regress' with 'degrade' to avoid confusion everywhere.
> > >
> > > **Q9: Why is DKM sometimes referred to as an "activation layer" (e.g., right above Section 3.2). This also confused me since there are weights involved, not activations.**
> > >
> > > **A9:** Thank you for the question. Weight is stored in the original layer, and DKM has no learnable parameter and works as an activation on top of weights. The DKM can be applied to output of a layer as well because it is an activation in essence.
> > > Here is a pseudo code (in pytorch style) for Conv2D for clarification.
> > > ```
> > > def forward(x):
> > >  clustered_weights = self.DKM(...)(self.weight)
> > >  return f.conv2d(x, clustered_weights)
> > > ```

---

> > > > ### Comment · Reviewer_JTKT · 2021-11-29
> > > > **Response**
> > > >
> > > > I thank the authors for their rebuttal. Most of my questions have been satisfactorily answered and I remain with my score.
> > > >
> > > > I think the importance of using a pretrained model for initialization should be clearly stated in the paper (I do not think that this is a major drawback of the proposed method).
> > > >
> > > > I am not sure whether any function without trainable parameters appearing in a neural network should be called an activation. I think activations have a somewhat clear meaning as the numbers that are being computed from the inputs of the previous layer through a linear operation. At least I recommend the authors to avoid the term "activation layer" in this context to avoid confusion.

---

> > > > > ### Author Response · Authors · 2021-11-30
> > > > > **Response to the comments**
> > > > >
> > > > > We do appreciate your comments and are happy that your questions are mostly answered. Regarding your final comment on the activation layer, we will address your concern in the final draft by avoiding “activation layer” as suggested. Thank you again.

---

### Official Review · Reviewer_r68F · 2021-11-03

**Correctness:** 3
**Technical Novelty And Significance:** 2
**Empirical Novelty And Significance:** 2
**Recommendation:** 5
**Confidence:** 4

**Main Review:**

- Strength
- Experimental results are done on important and representative datasets.
- Easy to understand what the paper is doing.

- Weakness

- I think the paper didn't provide a scientific way of answering the research question. It reads to me that the main problem with existing method is that a discrete, 1 center clustering is the problem so they relax it. If this were true, then I believe the following 2 experiments should also be done:

 a) used a fixed ratio from 3 or 4 nearest clusters (each with ratio .33 , .25). This sort of fixed attention has been studied and used in accelerating transformer inference and reported good performance. If the hypothesis in this paper were right, by using this fixed attention should also give good results.

  b) also try to learn it with other relaxation method such as gumbel softmax to see if an improvement can be observed.

I think in general the method proposed in this paper is simple, which is ok to me. But honestly empirically I don't see a great improvement so I am more skeptical to the claim of the root cause. Only after other type of verification of this simple idea is provided I will be recommending this paper.

- Misc
Also I don't see the baseline methods I know is included. This paper mainly focused on model compression so speed is not a concern. Thus, I don't think the reason why hessian computation is a problem. I am not peculiarly familiar with quantization based method, but I believe Hawq-v3 (v2 is cited in this paper) is a competitive baseline method to discuss. I'd like to see authors add the experiments of this method and have some meaningful discussions. I will categorize this paper as quantization method so I think this comparison is important to gain more insights.

**Summary Of The Paper:**

The paper claims that a competitive branch of model compression method: Weight clustering, can be greatly improved by allowing a soft(or say differentiable) usage of clustering mechanism to achieve a better approximation result.

**Summary Of The Review:**

In sum, a very simple idea is proposed. I think simple is good, but empirical results don't quite convince me. I am seeking more ways of validating this idea in order to recommend it.

---

> ### Author Response · Authors · 2021-11-17
> **Responses to Reviewer1**
>
> Thanks for the feedbacks, and we were able to greatly improve our paper by addressing your comments especially on Gumbel-softmax. Model compression is a very challenging research area, especially in the regime of sub 4bit [1,2], and has significant impacts on model deployment in the real world [3]. Our method pushed the SOTA by a wide margin (4% top-1 accuracy improvement with the comparable compression ratio for Resnet18 on ImageNet1k). And, also our paper is the first one that demonstrated 64.3% top-1 accuracy for mobilenet-v1 with effective 1bit-per-weight.
>
> **Q0: used a fixed ratio from 3 or 4 nearest clusters (each with ratio .33 , .25). This sort of fixed attention has been studied and used in accelerating transformer inference and reported good performance. If the hypothesis in this paper were right, by using this fixed attention should also give good results.**
>
> **A0:** [cross reference **A2, A3, A0:** for Reviewer2] Thanks for the opportunity to clarify the key element in our method. Even though the early training would begin with attention like (0.33, 0.25, ...), it does NOT stay the same for long but will be closer to one-hot vector over time. Our method enforces the weight update to capture the clustering effect from DKM through forward/backward, and eventually tries to make the attention like (**0.95**, 0.003, 0.02,...) in the later training phase. Please look at Fig. 5 (b) where we can observe that the weight distribution is gradually getting clustered around two peaks (i.e., 1bit clustering), which also evolves the attention closer to a one-hot assignment. In detail, as weights are updated again and again, they get closer to one of the centroids over time which is shown in Fig. 7: the inference weights are discretely clustered for compression and accuracy thus each weight is exactly one of the centroids, while the train weights are cluster approximately using soft-assignment. Such difference creates the gap or error between inference vs. train weights, but such gaps get smaller over time (based on EM in Appendix G). Therefore, fixing the ratio will prevent such gradual clustering and won't perform as DKM is designed. We added the time direction to avoid confusion on how DKM helps weight-clustering to Fig. 5 (b).
>
> **Q1: also try to learn it with other relaxation method such as gumbel softmax to see if an improvement can be observed.**
>
> **A1:** Authors appreciate this critical point from the reviewer. Indeed, Gumbel-softmax can be another way of computing soft assignment, and comparing it with our method and hard-assignment cases would greatly strengthen the value of our work. Hence, we used Gumbel-softmax to generate the attention part and kept the iteration to reduce the variance (a problem pointed by [4]). Then, we obtained the following Top-1 accuracy results on the 3 DKM compression configurations with ResNet18 from Fig 4.
>
> |approach | cv:4/4,fc:8/4 | cv:6/8,fc:6/10 | cv:5/8,fc:8/4 |
> | --- | --- | --- | --- |
> | DKM | 66.7 | 65.7 | 65.1 |
> |Gumbel-softmax | 65.7 | 62.9 | 61.7 |
> |hard-assignment | 54.9 | 52.1 | 49.4 |
>
> Clearly, it shows that a soft-assignment strategy is better than a hard-assignment one. Gumbel-softmax shows decent accuracy but is still worse than the proposed method on all the tested cases: 3.4% down from the most compressed case, which is very like due to the variance [4]. More graphs and explanations are added to Appendix F.
>
> A different kind of relaxation technique for model compression is proposed in LR-Net [4], and we compared LR-Net  with DKM for ResNet18 and ImageNet1k. LR-Let delivered about 60% Top-1 accuracy with 32x compression, while DKM offered 65.7% with a similar compression ratio. Meta-Quant [5] is another relaxation/approximation method where it *learns* how to generate gradient (i.e., for a given gradient *g*, Meta-Quant(*g*) is used instead during a backward pass). When we compared Meta-Quant with DKM, we found that DKM offers a superior trade-off than Meta-Quant: it offered 60.3% Top-1 accuracy with 32x compression. In any case, both LR-Net and Meta-Quant are better that the hard-assignment strategy. We update Fig. 4 (a) accordingly to compare DKM with LR-Net and Meta-Quant (along with other methods from the literature).
>
> **Q2: Misc Also I don't see the baseline methods I know is included. This paper mainly focused on model compression so speed is not a concern. Thus, I don't think the reason why hessian computation is a problem.**
>
> **A2:** Thanks for the comment. Although it is correct that high-computation cost such as hessian trace is not directly related to the model compression, it does indirectly as model compression requires exploring accuracy-compression trade-off (like ResNet18 in Fig 4). A faster train-time compression enables a user to explore the trade-off better and pick the best compression configuration for the user's need. In that sense, we believe that faster training time with compression can be highly desirable for many scenarios.

---

> > ### Author Response · Authors · 2021-11-17
> > **Responses to Reviewer1 [continue]**
> >
> > **Q3: I am not peculiarly familiar with quantization based method, but I believe Hawq-v3 (v2 is cited in this paper) is a competitive baseline method to discuss. I'd like to see authors add the experiments of this method and have some meaningful discussions. I will categorize this paper as quantization method so I think this comparison is important to gain more insights.**
> >
> > **A3:** Thank you for suggesting a new baseline. We picked PROFIT (ECCV20) and EWGS (CVPR21) as our baseline because they both have the state-of-the-art results on the sub 4-bit quantizations. Since Hawq-v3 (NeurIPS20) is another state-of-the-art in quantization, we performed additional experiments for a comprehensive study. Hence, we modified the public code (https://github.com/Zhen-Dong/HAWQ) and ran the experiments on our 16GPU platform using the default parameters from the authors with 4,3,2,1 bits for weight bit and full precision for activation. We mainly modified bit_config.py to specify the bit width for each layer and preserve the core of Hawq-v3 implementaion for accurate comparison.
> >
> > Our results show that Hawq-v3 is not as competitive as the existing compression algorithms in the sub 4bit regime, perhaps because it optimizes other targets such as latency and HW complexity. Regarding Resnet18, we were able to get a better result with 4bit than paper (due to 32bit activation precision), a competitive 3bit result, but 2bit result is worse than our method and others. Mobilenet_v2 results seem not competitive either. The Resnet18 and Mobilenet_v2 results with DKM are much superior to Hawq_v3 results as in Table 2.
> > Based on the goal of Hawq-v3 and its results for the sub 4bit compression, we think PROFIT and EWGS have more compatible goals with our method thus can serve as strong and competitive baselines.
> >
> > | | Hawq_v3:resnet18 |   Hawq_v3:mobilenet_v2 | DKM:resnet18 |   DKM:mobilenet_v2 |
> > | --- | --- |   --- | --- |   --- |
> > |4bit| 72.5 |   36.8 |  | |
> > |3bit| 69.1 |  29.8 | 69.9 | 70.3|
> > |2bit| 24.2 |   not converging | 68.9 | 66.2|
> > |1bit| not converging   | not converging | 65.0 | 50.8|
> >
> > **Reference**
> >
> > [1] Eunhyeok Park and Sungjoo Yoo. Profit: A novel training method for sub-4-bit mobilenet models. In European Conference on Computer Vision, 2020.
> >
> > [2] B.Ham J. Lee, D. Kim. Network quantization with element-wise gradient scaling. In Proceedingsof the IEEE Conference on Computer Vision and Pattern Recognition, 2021
> >
> > [3] Kuan Wang, Zhijian Liu, Yujun Lin, Ji Lin, and Song Han. Haq: Hardware-aware automated quantization with mixed precision. InProceedings of the IEEE Conference on Computer Vision and Pattern Recognition, 2019b
> >
> > [4] Oran Shayer, Dan Levi, and Ethan Fetaya. Learning discrete weights using the local reparameterization trick. In International Conference on Learning Representations, 2018
> >
> > [5] Shangyu Chen, Wenya Wang, and Sinno Jialin Pan. Metaquant: Learning to quantize by learning to penetrate non-differentiable quantization. In Advances in Neural Information Processing Systems,2019.

---

> > > ### Comment · Reviewer_r68F · 2021-11-22
> > > **Additional Question**
> > >
> > > Hi authors,
> > >
> > > Thanks for the response! I have 2 additional question.
> > >
> > > 1) I think your reply A1 is a bit off my point. I don't think it's important to beat other methods as in reply A1. The important thing to see is to understand what's the real meat of the your proposed method. So showing your method is better than others doesn't quite do the work as I don't know your initialization and hyperparemeters. But it reads to me weird as if hard-assignment performed badly, that would mean clustering is not useful. And gumbel softmax mathematically contains your method. So I expect it works more or less the same around your results but it seems like it performs badly on some setups according to your result. I couldn't understand the point. Overall, the reply doesn't resolve the my confusion on why the proposed method works or what's the key feature that makes it work. Your explanation is the soft relaxation can it can only convince me if other type of soft-relaxation also works more or less well. But according to your response it's not the case. Apologies for this but I really don't think the proposed method has any significant mathematical function to make it way better than other relaxations.
> > >
> > >
> > > 2) You keep mentioning < 4 bit is difficult so does that mean your proposed method works no better than other methods for > 4 bits?
> > >
> > >
> > >
> > > 3) Thanks for your endeavor to bring HAQW in. In general I appreciate your hard work and extensive experiments. I will increase my rating 1 above first and discuss with other reviewer to see if I will eventually recommend the paper. I understand people in compression field in general only cares about the numbers but I do want to see the reasons more. And as I said I don't see any particularly interesting formulation in your method. Maybe you can also point out a single line in your method that without that the method won't work. Or you can also design experiments to convince me why the proposed relaxation is the best, which I don't see from mathematical point of view.

---

> > > > ### Author Response · Authors · 2021-11-23
> > > > **Response**
> > > >
> > > > Thank you for increasing your rating and for the follow-up questions.
> > > >
> > > > **Q4. The important thing to see is to understand what's the real meat of the your proposed method. So showing your method is better than others doesn't quite do the work as I don't know your initialization and hyperparemeters**
> > > >
> > > > **A4:** Our method is motivated by the examples in Fig 1., and addresses the concern very efficiently, as demonstrated by our empirical results (**considerably pushing the state-of-the-art**).
> > > >
> > > > Our method is theoretically connected to the Expectation-Maximization (or EM) in terms of formulation as discussed in Appendix G. However, we solve the EM using the back-propagation from the gradient of the loss to optimize the observations (or weight), rather than the centroids: the conventional EM iteratively finds centroids for given fixed observations. One way to understand the key difference is following:  in EM, observation is not a variable but a static constant (which is no longer valid for deep learning). Thus, in our DKM, we take the weights as optimizable variables, and cast it as an activation layer to get the clustered centroid as output. This is the very first time that an activation layer with clustering behavior is proposed to our best knowledge.
> > > >
> > > > **Q5. But it reads to me weird as if hard-assignment performed badly, that would mean clustering is not useful**
> > > >
> > > > **A5:** Thanks for the chance to clarify this. DeepCompression (Song Han et. al. ICLR2016) is the early pioneering work that applied hard-assignment in clustering for model compression and demonstrated good results on over-sized networks like AlexNet/VGG. DeepCompression has been referred to over 6000 times and impacted many industry products (for example, https://research.fb.com/wp-content/uploads/2018/12/Machine-Learning-at-Facebook-Understanding-Inference-at-the-Edge.pdf). We believe that model compression is a critical topic in deep learning and clustering is one of the most popular/promising approaches. Along that line, we hope our fining (soft assignment is better than hard assignment for weight clustering) and results (pushing the state-of-the-art on model compression) would encourage more research in this domain and further help the industry deploy more optimized networks for inference.
> > > >
> > > > **Q6. And Gumbel softmax mathematically contains your method. So I expect it works more or less the same around your results but it seems like it performs badly on some setups according to your result.**
> > > >
> > > > **A6:** Thanks for the point. In fact, our results show Gumbel softmax works well. It constantly beats hard assignment by a large margin (more than 10%), and also outperforms the state-of-the-art (ATB) as shown in Fig 4. (a). So, this confirms that soft assignment is the key, and Gumbel softmax is very effective, yet our method is more effective. The Gumbel softmax is a continuous distribution thus has variance, while ours does not. Such variance could make training confused, as it can sometimes  generate high attention to far-away centroids, while our method doesn't have variance and our attention can be much more consistent throughout training. Regarding our implementation, to ensure fairness, we use $a_{i,j}$ = gumbel_softmax($\frac{d_{i,j}}{\tau}$) with the same temperature $\tau$.
> > > >
> > > > Note that this downside with the Gumbel-softmax for discretization or clustering has been discussed in the prior art: please see Fig. 4 in "Oran Shayer, Dan Levi, and Ethan Fetaya. Learning discrete weights using the local reparameterization trick. In International Conference on Learning Representations, 2018".
> > > >
> > > > **Q7: You keep mentioning < 4 bit is difficult so does that mean your proposed method works no better than other methods for > 4 bits?**
> > > >
> > > > **A7:** Compression with 4bits or higher is easy, and recent methods including ours can get the same accuracy as the 32bit case (which is the maximum possible) for the popular networks in Section 4. In that sense, one can roughly state that most of methods in recent years will work equally good for the >=4bit configurations. Hence, many latest papers on compression focus on sub 4bit compression and hard-to-compress networks like MobileNet. Note that we demonstrated 63.9% top-1 ImageNet1k accuracy with 0.72 MB model size (22.4x model compression factor), which is the much superior state-of-the-art to the existing know results.

---

### Official Review · Reviewer_yNNv · 2021-11-03

**Correctness:** 3
**Technical Novelty And Significance:** 2
**Empirical Novelty And Significance:** 3
**Recommendation:** 6
**Confidence:** 4

**Main Review:**

The paper is well-written and easy to follow. Using an attention matrix to optimize cluster centers with updates from all the weights is a promising and, to the best of my knowledge, a novel idea. I like Figure 4-b since it shows very clearly that the weight distribution is trained to be centered around clusters as desired. Here are some of questions and concerns:

- The authors claim that weight cluster assignments have not been optimized in prior work. However, although the clusters assignments are not optimized directly through back propagation, there is still an iterative process to optimize the assignments in many model quantization works such as [1] and [2]. I believe making this distinction more clear would be fairer to the prior work.

-  If I understand it correctly, the proposed method DKM is applied to each layer separately with different weight cluster centers and attention matrices. It seems like one bottleneck for the proposed method is that the storage complexity increases a lot for deep neural networks. Do the authors have an idea of to what extent these cluster centers and/or attention matrices might be shared across layers? Also, the cluster centers and attention matrices change at every iteration (before doing a forward pass). Can the authors provide an approximate number for the number of iterations required before each forward pass (this corresponds to $r$ in the paper)?

- I am a bit confused about the third paragraph in Section 3.3. Multi-Dimensional DKM. The authors make a connection between entropy and model quality and they say "... its entropy is $b$.". I am not sure what "it" refers to here. I thought $2^b$ was the number of clusters. I do not think entropy is the right term to use here. Can the authors clarify what they mean in that paragraph?

[1] Choi, Yoojin, Mostafa El-Khamy, and Jungwon Lee. "Towards the limit of network quantization." arXiv preprint arXiv:1612.01543 (2016).

[2] Choi, Yoojin, Mostafa El-Khamy, and Jungwon Lee. "Universal deep neural network compression." IEEE Journal of Selected Topics in Signal Processing 14.4 (2020): 715-726.

**Summary Of The Paper:**

This paper studies k-means clustering for deep neural network (DNN) compression with several improvements over prior work. The authors provide a differentiable k-means clustering layer (DKM) to optimize the cluster centers and cluster assignments throughout the training without separating the problem of training and compression.

**Summary Of The Review:**

The authors propose a simple but novel clustering method for DNN compression. Overall, I believe it has useful findings but also has some flaws that I mentioned in the previous section. Hence, my score is 6: marginally above the acceptance threshold.

---

> ### Author Response · Authors · 2021-11-17
> **Responses to Reviewer0**
>
> First, thank you for your valuable reviews. We have addressed all of your feedbacks in the paper. Here we provide our responses. Please refer to the paper for greater details.
>
> **Q0: The authors claim that weight cluster assignments have not been optimized in prior work. However, although the clusters assignments are not optimized directly through back propagation, there is still an iterative process to optimize the assignments in many model quantization works such as [1] and [2]. I believe making this distinction more clear would be fairer to the prior work.**
>
> **A0:** Thanks for bringing these papers to our attention. We referred to these two papers, modified the claims in Section 3.1, and added the following discussions to make the distinction clear and fair in Section 2: [1,2] are optimizing the clusters to minimize the change in the loss by considering gradient and hessian of loss function at each iteration of training. Therefore, it is to preserve the current model state. However, in our method, we let gradients go through the attention values to enable the parameters to learn directly from the data without assumptions in [1,2]. Accordingly, In DKM, the weights will change to explore a different and potentially better clustering. We adjusted our claim in the paper to reflect the above statements.
>
> **Q1: If I understand it correctly, the proposed method DKM is applied to each layer separately with different weight cluster centers and attention matrices. It seems like one bottleneck for the proposed method is that the storage complexity increases a lot for deep neural networks. Do the authors have an idea of to what extent these cluster centers and/or attention matrices might be shared across layers?**
>
> **A1:** Thanks for bringing up an essential direction for our future work. During train-time, the forward pass needs to keep a copy of the centroid and attention matrix, and the attention matrix size is proportional to the weight. If we share them across layers, we may save on the centroid but not on the attention matrix as attention needs to be computed for each (weight-centroid) pair. For example, let's think of two layers **L0**, **L1** with weights **w0**, **w1** and centroids **g0** ,**g1**.
>
> #### memory-overhead during training, when NOT shared
> * L0: |g0| from the lookup table + |w0||g0| from the attention.
> * L1: |g1| + |w1||g1|
>
> #### memory-overhead during training, when g0 and g1 shared as g*
> * |g*| >= |g0| = |g1|
> * L0 and L1 combined: |g*| + (|w0|+|w1|)|g*| --> **Still the same attention overhead exits**
>
> #### memory-overhead during inference when g0 and g1 shared as g*
> * L0 and L1 combined: |g*|
>
> Hence, we believe that the sparse representation of the attention matrix is the effective way of addressing the memory bottleneck as in our conclusion because the attention matrix will be sparse by nature (i.e., weights will have near-zero attention to the centroid far away).
>
> **Q2: the cluster centers and attention matrices change at every iteration (before doing a forward pass). Can the authors provide an approximate number for the number of iterations required before each forward pass (this corresponds to r in the paper)?**
>
> **A2:** [cross reference **A7:** for Reviewer2] Thanks for suggesting a vital experiment. We ran additional experiments to gather the number of iterations per layer for each batch with ResNet18 and plotted them in Fig. 8 in Appendix D. We can see that the number of iterations is high at the beginning of training (hitting the maximum limit of iterations 5), but becomes small after about 2500 batches (5-6 epochs). Also, we noticed that each layer shows a different trend, especially the final FC layer needs more iterations than other convolutional layers. We specified the maximum iteration limit in Section 4, and added more details in Appendix D.
>
> **Q3: I am a bit confused about the third paragraph in Section 3.3. Multi-Dimensional DKM. The authors make a connection between entropy and model quality and they say "... its entropy is b .". I am not sure what "it" refers to here. I thought 2 b  was the number of clusters. I do not think entropy is the right term to use here. Can the authors clarify what they mean in that paragraph?**
>
> **A3:** Thank you for the question. In fact, "it" refers to the weight matrix and its entropy is indeed 'b' which is the number of bits for clusters. We added Fig. 3 to Section 3.3 to increase the clarity. In detail, when we assume each centroid is used equal times in compressing the weight matrix, its probability of usage is $\frac{1}{2^b}$ and its entropy becomes $b$ as in Fig 3. And, it is shown that there is a connection between entropy and model quality (higher is better) (Park et. 2017). As an intuitive example, a model with 16bit precision ($b=16$) learns better than one with 2bit precision ($b=2$). We explain that multi-dim DKM   increases the entropy (thus model accuracy) with the same compression target, which is demonstrated in Table 2.

---

> > ### Comment · Reviewer_yNNv · 2021-11-22
> > **Thanks for the response!**
> >
> > I would like to thank the authors for the additional experimental results (Q2) and clarifications. Most of my concerns are resolved. I will finalize my score after I check the revised manuscript in more detail and discuss the paper with other reviewers.

---

> > > ### Author Response · Authors · 2021-11-23
> > > **Thank you**
> > >
> > > Thank you

---

> > > ### Comment · Reviewer_yNNv · 2021-12-06
> > > **Final score**
> > >
> > > I thank the authors again for addressing my concerns and revising the paper. I think the paper has useful results and I keep my original score.

---

### Author Response · Authors · 2021-11-17
**Summary of changes**

We thank all the reviewers for their thorough comments and their many suggestions for improving the exposition of the paper. We have updated the paper with the following main changes:

1. We revised Section 3.3 on multi-dim DKM for clarification and added a new figure for better understanding.
2. We measured the gap between train-time vs. inference-time weights and added a new table and graph for discussion in Section 4 and Appendix C.
3. We ran additional experiments to measure the number of iterations in our method and reported the results in Appendix D.
4. We evaluated the impact of various $\epsilon$ values and added a new figure for insights in Appendix E.
5. We ran additional experiments with the Gumbel-softmax approach and hard assignment for reliable comparison and added new results in Section 4 and Appendix F.
6. We explained our theoretical connection to the Expectation-Maximize method in Section 3 and Appendix G.
7. We fixed typos and cleaned up the reference format.

---

### Decision · Program_Chairs · 2022-01-20

**Decision:**

Accept (Poster)

**Comment:**

The paper proposes a simple approach to quantizing neural network weights with encouraging empirical results. The authors did work hard to improve the paper and address reviewers' concerns during the discussion period. I believe the presentation of results can improve by adding a discussion of inference time. I am not sure if all of the baselines (e.g., in Figure 4) have the same inference cost.

PS1: The method does seem to unroll the iterative optimization process (ie. EM) of a Gaussian mixture model (GMM) and differentiates through the unrolled iterations. The paper makes the connection to attention, but does not seem to make a clear connection with GMM and EM. If this connection is correct, adding a discussion can be helpful.

PS2: I am not a big fan of using differentiable k-means as the method name. Differentiable k-means is confusing partly because k-means is differentiable, i.e., one can optimize k-means centers using gradient descent. The proposed approach seems more relevant to meta-learning, where one differentiate though one optimization process to optimize a secondary objective.